# Learning conformational ensembles of proteins based on backbone geometry

**Nicolas Wolf** [*1,2,3]    **Leif Seute** [*2,1,3]    **Vsevolod Viliuga** [4,1]    **Simon Wagner** [3]
**Jan Stühmer** [2,5]    **Frauke Gräter** [1,2,3]

[1]Max Planck Institute for Polymer Research, Mainz, Germany
[2]Heidelberg Institute for Theoretical Studies, Heidelberg, Germany
[3]IWR, Heidelberg University, Heidelberg, Germany
[4]SciLifeLab and DBB at Stockholm University, Stockholm, Sweden
[5]IAR, Karlsruhe Institute of Technology, Karlsruhe, Germany

## Abstract

Deep generative models have recently been proposed for sampling protein conformations from the Boltzmann distribution, as an alternative to often prohibitively expensive Molecular Dynamics simulations. However, current state-of-the-art approaches rely on fine-tuning pre-trained folding models and evolutionary sequence information, limiting their applicability and efficiency, and introducing potential biases. In this work, we propose a flow matching model for sampling protein conformations based solely on backbone geometry – BBFlow. We introduce a geometric encoding of the backbone equilibrium structure as input and propose to condition not only the flow but also the prior distribution on the respective equilibrium structure, eliminating the need for evolutionary information. The resulting model is orders of magnitudes faster than current state-of-the-art approaches at comparable accuracy, is transferable to multi-chain proteins, and can be trained from scratch in a few GPU days. In our experiments, we demonstrate that the proposed model achieves competitive performance with reduced inference time, across not only an established benchmark of naturally occurring proteins but also *de novo* proteins, for which evolutionary information is scarce or absent. BBFlow is available at `https://github.com/graeter-group/bbflow`.

## 1 Introduction

In recent years, the field of protein structure prediction has been revolutionized by geometric deep learning [19, 4, 27]. Jumper et al. [19] introduced AlphaFold 2, which predicts a protein's structure using patterns found in naturally occurring protein sequences, so-called *evolutionary information*, upon inference. On the other hand, advancements in generative modeling such as diffusion [39] and flow-matching [29, 3, 41] have propelled the field of protein design, where several approaches for the generation of novel protein structures have been proposed [46, 52, 7]. Plausible protein structures conditioned on symmetry or a motif can be designed without requiring an input sequence [15, 14, 53].

Both, protein structure prediction and design methods, generate a single *equilibrium structure* of a protein. In contrast, protein function depends on structural dynamics [34, 11, 5], that is, the protein's conformational ensemble as given by the Boltzmann distribution, assuming thermal equilibrium. To sample from the Boltzmann distribution, Molecular Dynamics (MD) simulations are an established method in the field [2]. However, covering the state space extensively with MD requires long

---

*Equal contribution.

39th Conference on Neural Information Processing Systems (NeurIPS 2025).

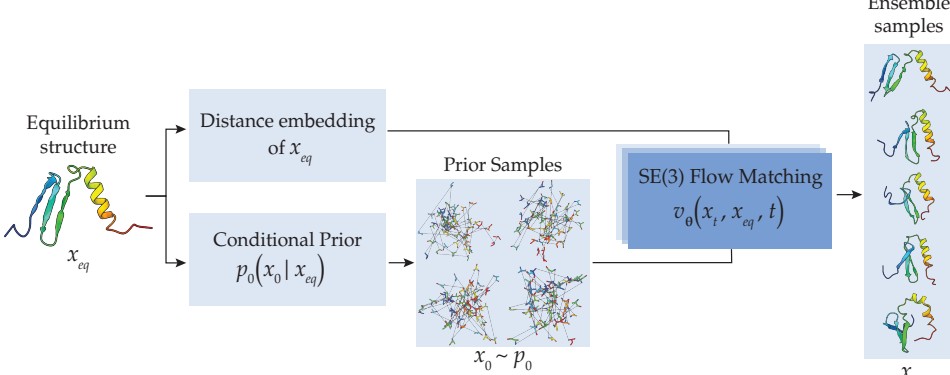

Figure 1: Schematic representation of BBFlow. The equilibrium backbone structure $x_{eq}$ of an input protein is used to condition an SE(3) Flow Matching model on the generation of protein backbone conformations $x_1$. Already the prior $p_0$ of the flow matching process is conditioned on the input protein via partial geodesic interpolation between pure noise and the equilibrium backbone structure.

simulation times in order to satisfy ergodicity by overcoming local free energy minima, making conformational sampling often prohibitively expensive. Recently, generative models have been suggested for emulating the sampling of MD conformations, offering inference times that are orders of magnitudes faster than MD [33].

For proteins, current state-of-the-art approaches for such generative models rely on modifications of AlphaFold 2, where noise is introduced into the MSA [47], the pre-trained folding model is fine-tuned on ensemble data [16], or the structure block is replaced by a diffusion model [24]. While these approaches are capable of generating realistic conformational ensembles, their efficiency is limited since they depend on large pre-trained folding models or the sampling of each state requires to predict the overall fold of the protein from the sequence. Consequently, the models rely on processing of evolutionary information such as MSA or weights from protein language models like ESM [27]. This renders the models not only expensive, but can in addition introduce biases for sequences where evolutionary information is scarce or, as for *de novo* proteins, absent [27].

**Main contributions**   In this work, we introduce BBFlow, a generative model for MD-derived protein conformational ensembles based on backbone geometry that is *more than an order of magnitude faster* than the current state-of-the-art model AlphaFlow [16], at similar accuracy. To the best of our knowledge, it is the first protein ensemble generation model shown to be applicable not only to monomeric but also to *multi-chain proteins*. BBFlow relies on two key innovations. **(1)** We formulate conformational ensemble prediction as protein structure generation task, conditioned on a geometric encoding of the equilibrium structure and **(2)** propose a conditional prior distribution for flow matching based on geodesic interpolation (Fig. 1). Notably, our work shows that neither pre-trained weights from a folding model nor evolutionary sequence information are necessarily required to generate conformational ensembles as observed in 300 nanoseconds (ns) of MD.

For benchmarking BBFlow, we train and test the model on the ATLAS dataset [42], which contains a curated set of 300 ns long Molecular Dynamics trajectories for 1390 proteins – the same dataset used for training AlphaFlow. We also test BBFlow on MD trajectories of *de novo* proteins, where we find similar performance as for naturally occurring proteins while AlphaFlow fails if the equilibrium structure is not provided as template. We show that BBFlow – although trained only on monomeric proteins – can generalize to multi-chain proteins, which are not covered by other baselines.

## 1.1   Related Work

**Ensemble generation**   Previous deep learning approaches for sampling conformational ensembles that apply invertible neural networks [33] or equivariant flow matching [21] usually require training on the specific system of interest. For proteins, a transferable model, AlphaFlow, has been recently proposed [16], relying on fine-tuning the pre-trained folding model AlphaFold 2. Li et al. [25] propose to speed up AlphaFlow upon inference time by only calling the evoformer once, however, the model is not publicly available and only a subset of metrics is reported.

**Generating standardized MD-emulating ensembles**    AlphaFlow is trained on the ATLAS dataset [42], which contains MD conformations obtained in a standardized setting – via three times 100 ns long simulations at the same temperature, force field and water model. Training models to generate such standardized ensembles, which we refer to as *MD emulation*, is important for evaluating their distributional accuracy quantitatively. This is because ensemble metrics strongly depend on the duration and temperature of the corresponding MD (see also Tab. A.3).

Next to AlphaFlow, also ConfDiff [45], a diffusion model that relies on a pre-trained sequence representation of AlphaFold 2, and MDGen [17], a model that generates ensembles of molecules in all-atom representation with consistent time evolution, are trained on the ATLAS dataset.

**Generating other ensembles**    There is also interest in generating protein ensembles that do not strictly follow a certain distribution induced by a fixed MD time and temperature but rather sampling general alternative states, such as BioEmu [24]. We discuss such models in App. A.4 and Tab. A.3.

## 2    Background

### 2.1    Flow Matching for protein structure generation

**Flow Matching**    In order to sample from a target distribution $p_1 : \mathcal{M} \to [0, 1]$ on the data domain $\mathcal{M}$, Lipman et al. [29] have proposed flow matching as generalization of diffusion models [39]. A learned flow $\phi : \mathcal{M} \times [0, 1] \to \mathcal{M}$ is used to transform samples $x_0 \sim p_0$ from a simple prior distribution $p_0$ to samples $\phi(x_0, 1)$ from the target distribution $p_1$. The key idea is to learn a time-dependent flow vector field

$$v(x, t) : \mathcal{M} \times [0, 1] \to \mathcal{T}_x \mathcal{M}, \quad (x, t) \mapsto v(x, t), \tag{1}$$

where $\mathcal{T}_x \mathcal{M}$ is the tangent space at point $x$. The flow $\phi_t \equiv \phi(\cdot, t)$ is then defined by $v_t$ via integration of the flow ODE,

$$\frac{\mathrm{d}}{\mathrm{d}t} \phi_t(x) = v(\phi_t(x), t), \quad \phi_0(x) = x. \tag{2}$$

The vector field $v_t$ can be learned by sampling $x_0 \sim p_0$ and $x_1 \sim p_1$, connecting them by a particle-wise flow $\psi(x_0, x_1, t)$ and regressing on the time derivative of $\psi$ [29]. On Riemannian manifolds, $\psi$ is usually chosen as geodesic [9].

**Application to protein structure**    A protein backbone can be represented as a sequence of Euclidean frames $x = (r, z) \in \mathrm{SE}(3)$ [19], each of which consists of a rotation $r \in \mathrm{SO}(3)$ and a translation $z \in \mathbb{R}^3$. A flow matching process for protein structure can thus be formulated on the Riemannian manifold $\mathcal{M} \equiv \mathrm{SE}(3)^N$. By choosing the metric on $\mathrm{SE}(3)^N$ as in [51], the geodesic paths can be split into independent rotation and translation parts for each residue. Typically, one parametrizes both the ground truth and predicted vector field by a current structure $x_t$ and a final structure $x_1$. It can be shown [51, 7] that the vector field components are then given by

$$v_{\mathrm{SO}(3)}(r_t, t | r_1) = \frac{\log_{r_t}(r_1)}{1 - t}, \quad v_{\mathbb{R}^3}(z_t, t | z_1) = \frac{z_1 - z_t}{1 - t}. \tag{3}$$

A common choice for the prior distribution $p_0$ is independent Gaussians for the translations $z_0 \sim \mathcal{N}(0, \sigma^2)$ and uniform distributions for the rotations $r_0 \sim \mathcal{U}(\mathrm{SO}(3))$ [51].

### 2.2    Evolutionary sequence information

In order to determine the structure of a protein, the challenging task of mapping from a one-dimensional sequence representation to a three-dimensional backbone geometry needs to be solved. To achieve this, folding models like AlphaFold 2 rely on evolutionary information in the form of Multiple Sequence Alignment (MSA) – an algorithm that aligns the input sequence with related naturally occurring protein sequences from a database during inference and training to identify patterns that encode information on folding states such as pairwise contacts. The calculation of an MSA during inference is computationally expensive. A more efficient strategy is to encode evolutionary information by extracting weights from a protein language model [36], which can be seen as learned evolutionary information [27]. While evolutionary information has been shown to be beneficial

for predicting ensembles or alternative folding states [47], there is rising interest for methods that perform well also without relying on evolutionary information [12, 54], which is not available for de-novo proteins or the disordered proteome.

## 3 Method

In this work, we propose to decouple protein conformational ensemble generation from the structure prediction task and introduce a generative model based purely on backbone geometry that does not rely on evolutionary sequence information. We achieve this by conditioning both the flow and the prior on the equilibrium structure of the protein.

**Conditional flow matching for ensemble generation**    Inspired by FrameFlow [51], a flow matching model for protein structure design, we formulate MD emulation as structure generation task, conditioned on the equilibrium state of the respective protein. In particular, we express the Boltzmann distribution of a given protein as probability distribution $p(x|x_{\text{eq}})$ of conformations $x$, conditioned on the equilibrium state $x_{\text{eq}}$ of the respective protein. In order to sample from $p(x|x_{\text{eq}})$, we learn a flow vector field,

$$v(x, t, x_{\text{eq}}) : \mathcal{M} \times [0, 1] \times \mathcal{M}_{\text{eq}} \to T_x \mathcal{M} \,, \tag{4}$$

that receives protein equilibrium states $x_{\text{eq}} \in \mathcal{M}_{\text{eq}}$ as additional input. This defines a conditional flow $\phi_t$ by

$$\frac{\mathrm{d}}{\mathrm{d}t} \phi_t(x|x_{\text{eq}}) = v\left(\phi_t, t, x_{\text{eq}}\right), \quad \phi_0(x|x_{\text{eq}}) = x \,. \tag{5}$$

Crucially, by conditioning the generation not on the sequence but the equilibrium structure, we eliminate the need for evolutionary information and pre-trained folding model weights. We summarize the training procedure in Algorithm 1.

We note that assuming the availability of an equilibrium structure is reasonable because, as MD emulator, the use-case of the model is to offer an alternative to MD simulation, which also requires an initial structure (see A.1). If only a sequence is available, both MD and BBFlow first require a structure prediction with a folding model. In Tab. A.1, we show that BBFlow remains accurate and fast if used as sequence-to-ensemble model in combination with AlphaFold 2.

**Model architecture**    In order to learn the conditional flow vector field $v_t$, we adapt the model architecture of the recent protein design model GAFL [44], which is an extension of the FrameDiff architecture proposed by Yim et al. [52]. The input features include the frames $x_t$ at time $t$, their pairwise spatial distances, and the flow matching time $t$. Crucially, in contrast to common protein structure architectures [52, 51, 19, 16], we do not use the residue indices as input feature. The reasoning behind this choice is that the ordering of the residues in the chain is already encoded geometrically in the equilibrium structure. Removing the residue index as an input feature reduces memorization and enables transferability from monomers to multi-chain proteins as explained in Sec. 4.3.

The neural network is an SE(3) equivariant graph neural network, which uses invariant point attention (IPA) [19] as core element. In GAFL, IPA is extended to Clifford frame attention (CFA), where geometric features are represented in the projective geometric algebra and messages are constructed using the bilinear products of the algebra. Frames are consecutively updated along with node and edge features in a series of 6 message passing blocks to predict the target frames $x_1$. Compared to Alphafold 2 [19], this architecture is more shallow and operates only on structural data, hence a sequence-processing module like the Evoformer of AlphaFold 2 is not required.

**Encoding of the equilibrium structure**    For conditioning the flow vector field as in Eq. 4, we modify the architecture such that the equilibrium backbone structure of the protein can be used as input feature. Inspired by the interpretation of evolutionary information as contact map [27], we propose to encode pairwise distances of the equilibrium state $x_{\text{eq}}$ as initial edge feature,

$$s_{ij} \equiv \text{bin}\left(||z_i - z_j||_2\right) \,, \tag{6}$$

where we bin the distance uniformly between 0 and 20Å with bin count 22 [51]. Additionally, we encode the equilibrium structure in a more direct, geometrically meaningful way. Inspired by

tensor-based equivariant networks [38] and their formulation in terms of local frames [30], we include equivariant pairwise directions between residues that are closer than 5Å as unit vectors,

$$e_{ij} \equiv r_i^{-1} \left( \frac{z_i - z_j}{||z_i - z_j||_2} \right),$$  (7)

and express them in the coordinate frame $x_i = (r_i, z_i)$ of residue $i$. Through the transformation into the co-rotating coordinate frame, the feature components become invariant and can be used together with $s_{ij}$ as initial edge feature.

We use amino acid identities as additional node features by transforming a one-hot encoding via a linear layer to a 128-dimensional embedding. The reasoning behind encoding the amino acid type is that it carries information about the local degrees of freedom of the backbone, however, in an ablation (Tab. 3) we find that also without the amino acid identity, the model performs remarkably well.

**Conditional prior distribution**   Unlike diffusion models [39], where Gaussianity of the prior $p_0$ is a strict theoretical requirement, flow matching, in principle, allows the use of general prior distributions [29]. Non-Gaussian, unconditional prior distributions for proteins have been proposed by Ingraham et al. [15] and Jing et al. [16]. We take this idea a step further and propose a *conditional* prior distribution $p_0(x|x_{\text{eq}})$ for flow matching. Samples $x_0 \sim p_0(\cdot|x_{\text{eq}})$ are generated by interpolating between samples from an unconditional prior $p_{\text{uncond}}$ and the equilibrium structure $x_{\text{eq}}$,

$$x_{\text{uncond}} \sim p_{\text{uncond}}, \quad x_0 \equiv \gamma(x_{\text{uncond}}, x_{\text{eq}}, \xi),$$  (8)

where $\gamma$ is the geodesic between $x_{\text{uncond}}$ and $x_{\text{eq}}$,

$$\gamma(x_{\text{uncond}}, x_{\text{eq}}, 0) = x_{\text{uncond}}, \quad \gamma(x_{\text{uncond}}, x_{\text{eq}}, 1) = x_{\text{eq}},$$  (9)

and $\xi$ is a hyperparameter between 0 and 1 that quantifies how close the noise sampled from the prior is to the equilibrium structure (see Fig. A.4). In our experiments, we set $\xi \equiv 0.2$. For an ablation of $\xi$, see Sec. A.8. For the unconditional prior distribution $p_{\text{uncond}}$, we use the normal distribution for translations and the uniform distribution for rotations [51]. We note that this approach of conditioning the prior can be seen as generalization of partial denoising from diffusion [31] to the flow matching framework.

**Loss function**   As explained in Sec. 2, we represent protein backbone structure as a set of frames $x = (r, z) \in \text{SE}(3)$ and define the flow matching process on the data manifold $\mathcal{M} \equiv \text{SE}(3)^N$. We learn a conditional flow vector field $\hat{v}(x_t, t, x_{\text{eq}})$ (Eq. 4) on the tangent space of the data domain, parametrized by Eq. 3. For regressing on this vector field, we calculate the ground truth $v$ as tangent vector to the geodesic $\gamma_{\text{FM}}$ between the prior sample $x_0$ and target sample $x_1$, and apply a mean squared error loss,

$$\mathcal{L}_{\text{FM}} = \mathbb{E}\left[ \left\| v - \hat{v}(x_t, t, x_{\text{eq}}) \right\|_{\text{SE}(3)}^2 \right],$$  (10)

where $x_t$ is a point along the geodesic $\gamma_{\text{FM}}$, $x_t \equiv \gamma_{\text{FM}}(x_0, x_1, t)$, and $x_{\text{eq}}$ denotes the equilibrium structure used as condition. The expectation in Eq. 10 runs over

$$t \sim \mathcal{U}(0, 1), \quad (x_1, x_{\text{eq}}) \sim p_{\text{data}}, \quad x_0 \sim p_0(\cdot|x_{\text{eq}}),$$  (11)

and the metric is defined as in [52],

$$\left\| v \right\|_{\text{SE}(3)}^2 \equiv \text{Tr}\left( v_r v_r^T \right)/2 + \left\| v_z \right\|_2^2,$$  (12)

with the Euclidean 2-norm $\| \cdot \|_2$ and the projection on rotational and translational subspaces $v = (v_r, v_z)$. As in [51], we also use the auxiliary loss proposed in [52].

## 4   Experiments

**Training**   In order to directly compare the proposed model to the current state-of-the-art MD emulator for proteins, AlphaFlow [16], we train BBFlow on the ATLAS dataset [42] with the same split into training, validation and test proteins. The ATLAS dataset consists of three trajectories of 100 ns long all-atom Molecular Dynamics (MD) simulations for 1390 structurally diverse proteins, of which Jing et al. [16] select 1265 for training, 39 for validation and 82 for testing. We train the

Table 1: Performance of BBFlow and baselines (Sec. 4) on the ATLAS test set. For each protein, we evaluate the metrics described in Sec. 4 and report the median of all proteins. We also report RMSF medians over all residues and indicate the MD reference value in parentheses. Inference time is reported per generated conformation of the 302 residue protein 7c45A. All metrics except for correlations $r$ and transient contact accuracy $J_{\text{tr}}$ are reported in Å. Errors are estimated as described in Sec. 4 and are shown in parentheses if they are above precision. Best values are **bold**, second best are underlined. Note that BioEmu cannot be compared to other baselines directly, as explained in the paragraph **Further Baselines** below.

| | RMSF | | | Pw-RMSD | DCCM | PCA | $J_{\text{tr}}$ | Time |
|---|---|---|---|---|---|---|---|---|
| | $r$ (↑) | MAE (↓) | Median (MD=1.48) | MAE (↓) | $r$ (↑) | $\mathcal{W}_2$ (↓) | % (↑) | [s] (↓) |
| BioEmu* | 0.83 | 1.29 (0.01) | 2.34 | 2.84 (0.01) | 0.80 | 1.65 (0.04) | 36 | 1.9 |
| AlphaFlow | 0.86 | 0.59 (0.01) | 1.51 | 1.35 (0.01) | 0.86 | 1.47 (0.03) | 41 | 32.0 |
| ConfDiff | 0.88 | 0.62 (0.01) | 2.00 | 1.45 (0.01) | 0.86 | 1.41 (0.03) | 39 | 20.2 |
| AlphaFlow-T | **0.92** | **0.41** (0.01) | 1.17 | 0.91 (0.01) | **0.89** | **1.28** (0.03) | **47** | 32.6 |
| ESMFlow-T | **0.92** | 0.52 (0.01) | 0.94 | 1.22 (0.01) | **0.89** | 1.48 (0.03) | **47** | 11.2 |
| AlphaFlow-T$_{\text{dist}}$ | **0.92** | 0.68 (0.01) | 0.90 | 1.41 (0.01) | 0.88 | 1.43 (0.03) | 42 | 3.3 |
| AlphaFlow-T$_{\text{12L,dist}}$ | 0.90 | 0.85 (0.01) | 0.68 | 1.80 (0.01) | 0.87 | 1.60 (0.04) | 24 | 1.2 |
| BBFlow | 0.90 | 0.42 (0.01) | **1.49** | **0.77** (0.01) | 0.87 | 1.33 (0.03) | 29 | **0.8** |

*Not trained to generate ATLAS-ensembles.

model, and variants where we leave out key features for an ablation study, for 3 days on two NVIDIA A100-40GB GPUs from scratch, i.e. without initial weights from a pre-trained folding model. For all experiments, we use the same hyperparameters as in FrameFlow [51] and GAFL [44], except for the number of timesteps, which we set to 20. Also the respective feature dimensions are increased by 128 for embedding the amino acid identity as node feature and by 22 or 25, respectively, for embedding the equilibrium structure encoding with or without direction as edge feature.

**Baselines** We compare BBFlow with models from [16] that were fine-tuned on the training set of BBFlow, but rely on pre-trained weights from the folding models AlphaFold 2 and ESMFold [27] that were trained on much larger datasets. Next to the original AlphaFlow-MD model (referred to in this work as **AlphaFlow**), we also evaluate AlphaFlow-MD with templates (**AlphaFlow-T**), which receives the equilibrium structure as input, encoded as template in AlphaFold. Jing et al. [16] also introduce a model that relies not directly on the expensive MSA but on the protein language model ESM (**ESMFlow-T**). Additionally, we compare BBFlow with models based on AlphaFlow-MD with templates that are optimized for efficiency by distillation (**AlphaFlow-T$_{\text{dist}}$**), decreasing the timesteps required from 10 to 1, and by reducing the number of layers (**AlphaFlow-T$_{\text{12L,dist}}$**). We evaluate all models above using the conformations deposited in the AlphaFlow GitHub repository[2]. We also include the diffusion model **ConfDiff** [45] in our comparison (see Sec. 1.1).

**Further Baselines** For completeness, and because of their recent impact on the field, we also evaluate models that were not trained on the ATLAS dataset but on static structures, NMR data or longer MD trajectories, such as **BioEmu** [24] and other baselines [31, 32] in the appendix (Tab. A.3). These models are not trained to sample from the probability distribution of states visited during three times 100ns of MD and perform unfavorable if quantitatively evaluated in a standardized setting, as on the ATLAS test set (see Sec. 1.1). Further, we show in Tab. A.3 that BBFlow outperforms MDGen [17], an all-atom approach for generating time-consistent ensembles trained on ATLAS. For a comparison with the MSA subsampling approach [47] and the classical normal mode analysis (NMA) [8], we refer to [16], where it is shown that both perform worse than AlphaFlow and BBFlow.

**Metrics** We evaluate the performance of the compared models by reporting metrics that quantify how well statistical properties of the generated ensembles agree with those obtained by MD under standardized settings as described in Sec. 1.1. We report the key metrics, measuring similarity of the ensemble properties, Root Mean Square Fluctuation (RMSF), pairwise RMSD, principal

---
[2]https://github.com/bjing2016/alphaflow

components (PCA) and the Dynamical Cross Correlation Matrix (DCCM) that are established in the field of protein dynamics. To this end, we calculate the residue-wise Pearson correlation $r$ for RMSF and DCCM, the mean absolute error (MAE) for RMSF and pairwise RMSD and the Wasserstein-2 distance of the ensembles, projected on the first two principal components obtained from MD. To assess the role of pairwise distances in the provided equilibrium structure, we also report the accuracy of predicted transient contacts $J_{tr}$ as the Jaccard similarity between the generated ensemble and the MD ensemble. As in [16], we define a transient contact as a pair of residues that are separated in the equilibrium structure, but in proximity in 10% of the ensemble states, using a $C_\alpha$ distance threshold of 8 Å. In addition to these, Jing et al. [16] include new metrics, which we also report in an exhaustive evaluation table in the appendix (Tab. A.9). Furthermore, we investigate Ramachandran dihedral distributions in Sec. A.9. More details on the metrics and their interpretation can be found in Sec. A.3. In all experiments, we generate 250 conformations per protein, as in AlphaFlow, and bootstrap the set of MD conformations 100 times in order to estimate the error caused by sampling finitely many states. All metrics are calculated using the $C\alpha$ atoms of the protein structures.

**Inference time**  Inference time is, even if orders of magnitude smaller than MD, a critical factor for applications of MD emulators such as annotation of datasets or screening of proteins for a target motion. As in [14], we evaluate the inference time per generated conformation of the 302-residue protein 7c45A, and on the entire ATLAS test set in Fig. 5, using an NVIDIA A100-80GB GPU. Note that for models based on AlphaFold2, the inference time for all-atom structures is dominated by backbone generation (Tab. A.8).

## 4.1 ATLAS benchmark

We report the performance of BBFlow and the baselines evaluated on the ATLAS test set from AlphaFlow [16] in Tab. 1. We find that BBFlow generates high-quality conformational ensembles faster than all baselines. For proteins of length 300, it is around 40 times faster than AlphaFlow with templates (AlphaFlow-T), at comparable accuracy. While AlphaFlow-T is slightly more accurate in terms of RMSF and principal components, BBFlow outperforms it in capturing flexibility quantified by pairwise RMSD and median RMSF. BBFlow outperforms AlphaFlow, ESMFlow-T, the two distilled models and ConfDiff in almost all metrics while, at the same time, generating the ensembles faster. Indicated by small median RMSF, AlphaFlow-T systematically over-stabilizes the proteins and samples too close to the equilibrium structure. Additionally, we investigate the performance for different protein lengths (Fig. 2, Fig. A.6) and find that, while the trends described above hold true for all lengths considered, BBFlow performs favorably for larger proteins. At transient contact accuracy, BBFlow underperforms the baselines, indicating that for predicting rare events, evolutionary information might be required. For weak contacts (see A.3, Tab. A.9), BBFlow is more competitive.

Table 2: Performance of BBFlow and baselines for *de novo* proteins. Evaluation settings as in Tab. 1. Errors shown in parentheses if above precision. Best values are **bold**, second best are underlined.

| | RMSF | | | Pw-RMSD | DCCM | PCA | $J_{tr}$ | Time |
|---|---|---|---|---|---|---|---|---|
| | $r$ ($\uparrow$) | MAE ($\downarrow$) | Median (MD=0.91) | MAE ($\downarrow$) | $r$ ($\uparrow$) | $\mathcal{W}_2$ ($\downarrow$) | % ($\uparrow$) | [s] ($\downarrow$) |
| BioEmu* | 0.60 | 4.24 (0.01) | 7.56 | 8.29 | 0.64 | 1.53 (0.04) | 23 | 1.9 |
| AlphaFlow | 0.47 | 4.76 (0.01) | 7.09 | 7.40 | 0.58 | 1.64 (0.03) | 17 | 32.0 |
| ConfDiff | 0.62 | 3.82 (0.01) | 6.35 | 7.26 | 0.65 | 1.72 (0.02) | 15 | 20.2 |
| AlphaFlow-T | **0.89** | **0.25** (0.01) | 0.74 | 0.38 | 0.85 | 0.66 (0.01) | 55 | 32.6 |
| ESMFlow-T | **0.89** | 0.28 (0.01) | 0.68 | 0.43 | **0.86** | **0.63** (0.01) | **56** | 11.2 |
| AlphaFlow-T$_{dist}$ | 0.88 | 0.46 (0.01) | 0.51 | 0.77 | 0.84 | 0.69 (0.01) | 51 | 3.3 |
| AlphaFlow-T$_{12L,dist}$ | 0.87 | 0.58 (0.01) | 0.41 | 0.97 | 0.83 | 0.75 (0.01) | 38 | 1.2 |
| BBFlow | 0.84 | 0.26 (0.01) | **0.87** | **0.32** | 0.83 | 0.67 (0.01) | 32 | **0.8** |

*Not trained to generate ATLAS-ensembles.

## 4.2 De novo proteins

Recently, designing dynamical properties into novel proteins has gained attention [43, 22]. We hypothesize that BBFlow's greatly reduced inference time for generating high-quality ensembles

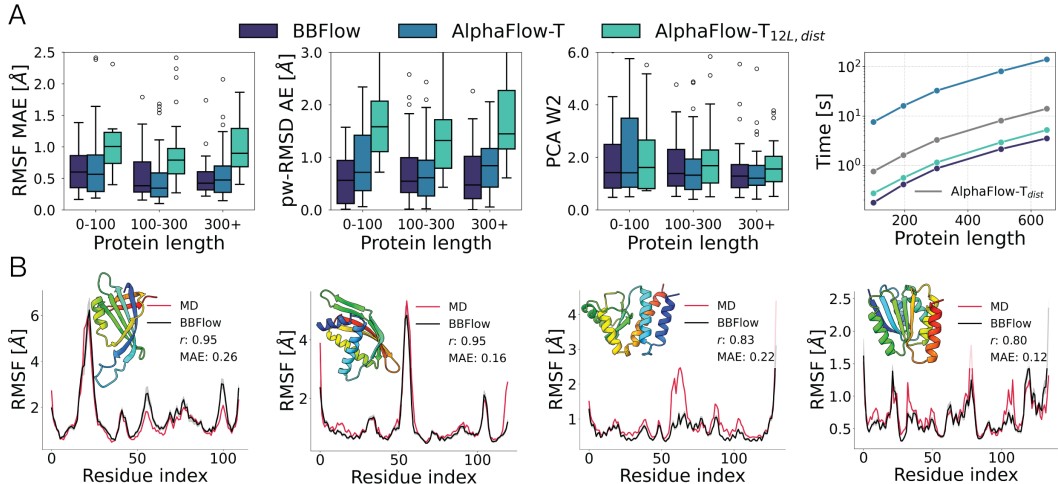

Figure 2: **(A)** Performance of BBFlow, AlphaFlow-T and AlphaFlow-T$_{12L,dist}$ on the ATLAS test set for different protein lengths. We divide the protein lengths in three bins and calculate RMSF MAE, the absolute error of pairwise RMSD and PCA $\mathcal{W}_2$ of each protein (lower is better) with length in the respective bin. The boxes depict the 0.25 and 0.75 quantile, minimum, maximum and median of all test proteins. We also show inference time per generated conformation as function of protein length in log-scale, spanning several orders of magnitude. **(B)** RMSF profiles of *de novo* proteins. We show structures and RMSF profiles predicted by BBFlow and MD of four selected proteins from the dataset of *de novo* proteins along with Pearson correlation $r$ and MAE as reported in Tab. 2.

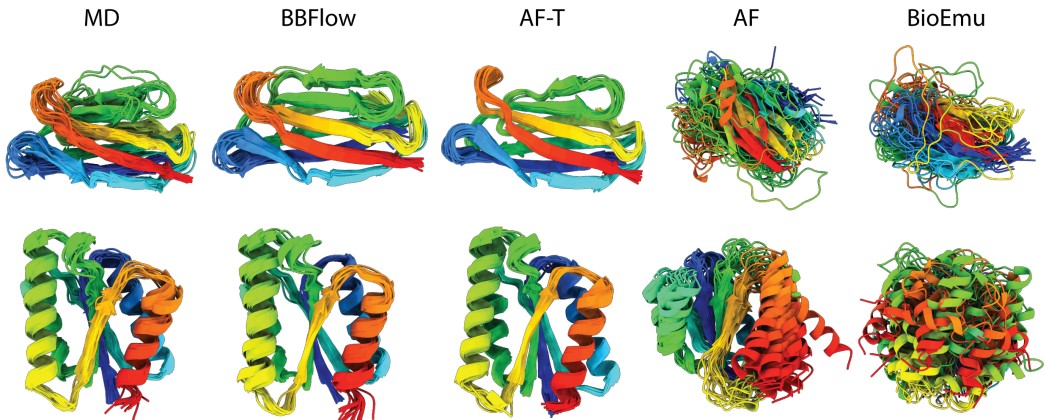

Figure 3: Ensembles of two *de novo* proteins predicted by BBFlow, AlphaFlow-T (AF-T), AlphaFlow (AF) and BioEmu compared with the ground truth molecular dynamics (MD) simulation. The proteins were generated by RFDiffusion and ProteinMPNN, and are colored by residue index.

makes the method interesting for applications in protein design pipelines, where efficient MD emulation would allow to screen for dynamics. However, since *de novo* proteins often have no evolutionary information available, the applicability of models that rely on such information is unclear. In order to evaluate conformational ensembles of *de novo* proteins, we generate a small dataset of 50 proteins sampled with the established models RFdiffusion [46] and FrameFlow [53], respectively, and perform three 100-ns-long MD simulations for each, similar to ATLAS (A.10).

In Tab. 2, we report the performance of the models considered in Sec. 4.1 for *de novo* proteins. We find that AlphaFlow without templates and BioEmu, which both heavily rely on evolutionary information, experience a strong decline of performance compared to naturally occurring proteins (Tab. 1). The relative differences between BBFlow and the other baselines are comparable to the performance on natural proteins. Fig. 2B displays structures and predicted RMSF profiles of four *de*

*novo* proteins. We also visualize ensembles of two randomly chosen *de novo* proteins predicted by BBFlow, AlphaFlow-T, AlphaFlow and BioEmu. Both AlphaFlow without templates and BioEmu fail to sample ensembles consistent with MD (Fig. 3), and instead tend to predict unstable conformations.

## 4.3 Multi-chain proteins

In contrast to folding architectures [19, 16, 52], BBFlow does not rely on absolute residue or chain indices as input features but rather on geometric biases imposed by the distance matrix of the equilibrium structure (see Sec. 3). It thus naturally extends to protein complexes that are made up of several protein chains, which are not covered by the baselines. This makes BBFlow, to the best of our knowledge, the first ensemble generation model shown to be applicable to multi-chain proteins.

We demonstrate in Tab. A.4 that BBFlow, indeed, captures ensemble properties of five well studied multi-chain systems (see App. A.5) and show that AlphaFlow, given multi-chain features as input, fails (App. A.5 and Fig. A.1). We visualize multi-chain ensembles and their DCCM computed with MD and BBFlow in Fig. 4 to illustrate that both intra-chain and inter-chain motions are captured by the model. Remarkably, it is able to do so without being trained on multimeric proteins but only on the single-chain proteins from the ATLAS dataset.

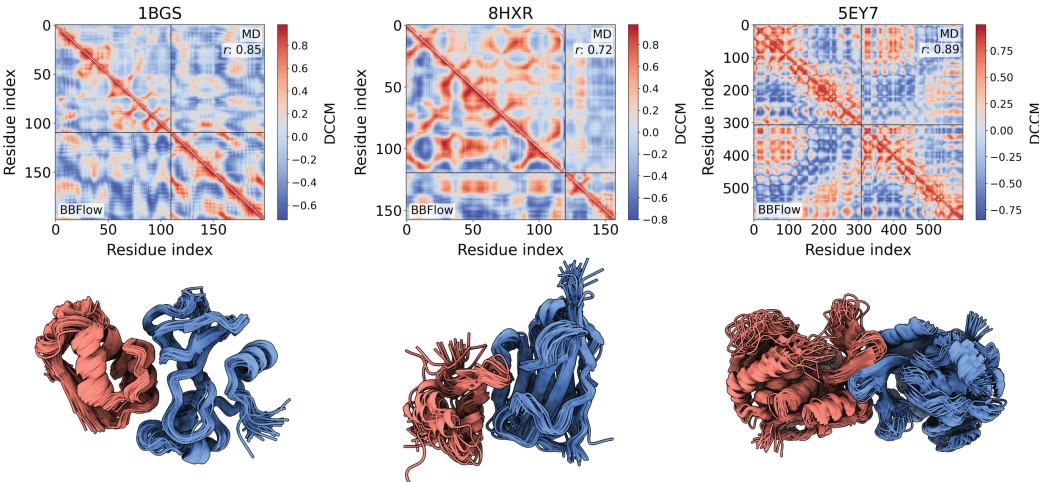

Figure 4: BBlow is applicable to multi-chain proteins. Dynamic cross-correlation matrices (DCCM) of conformational ensembles computed either with MD (upper triangle) or with BBFlow (lower triangle) for three protein dimers. Chain boundaries are indicated by black lines within matrices. $r$: Pearson correlation between entries of the triangular matrices. We show RMSF profiles in Fig. A.5.

Table 3: Ablation study for key components of BBFlow. Metrics are reported as in Tab. 1. Errors are calculated as described in Sec. 4 and displayed in parentheses if above precision.

| Name | Cond. prior | Distance encoding | Direction encoding | Amino acid enc. | Index encoding | RMSF MAE (↓) | Pw-RMSD MAE (↓) | DCCM $r$ (↑) |
|---|---|---|---|---|---|---|---|---|
| BBFlow | ✓ | ✓ | ✓ | ✓ | | **0.42**(0.01) | **0.77** (0.01) | 0.87 |
| a | | ✓ | | ✓ | ✓ | 0.52 (0.01) | 1.15 (0.01) | 0.85 |
| b | | ✓ | ✓ | ✓ | ✓ | 0.48 (0.01) | 0.90 (0.01) | 0.86 |
| c | ✓ | ✓ | ✓ | ✓ | ✓ | **0.42** (0.01) | 0.82 (0.01) | **0.88** |
| d | ✓ | ✓ | | | ✓ | 0.54 (0.01) | 0.93 (0.01) | 0.85 |
| e | ✓ | | | ✓ | ✓ | 5.88 (0.01) | 7.08 (0.01) | 0.55 |

## 4.4 Ablation

For quantifying the contributions of key components proposed or discussed in this work, we perform an ablation study on the ATLAS dataset and report the results in Tab. 3. We find that using the proposed direction encoding (a), the novel conditional prior (b) and eliminating the index encoding (c) indeed benefits the performance of the model. Additionally, we train a model that is entirely backbone structure-based, without any sequence information (d), and find that it is on par with non-template AlphaFlow. We also demonstrate the need for the distance encoding of the equilibrium structure (e).

## 4.5 Discussion

The results show that BBFlow achieves state-of-the-art performance in the trade-off between speed and quality of the generated ensembles (see also Fig. 5). At comparable accuracy, it is more than an order of magnitude faster than the current state-of-the-art model AlphaFlow-T and also faster than the distilled model AlphaFlow-T$_{12L,dist}$. Crucially, BBFlow does not suffer from the over-stabilization observed in AlphaFlow-T, impeding the exploration of conformational space. This performance is due to the proposed conditional prior and geometric encoding of the equilibrium structure, as shown in our ablation.

Not using templates in AlphaFlow can avoid overstabilization, but causes AlphaFlow to fail for *de novo* proteins. As a consequence, BBFlow is the only model considered that accurately captures overall flexibility for *de novo* proteins. We attribute these observations to BBFlow being based entirely on backbone geometry instead of evolutionary information, which is scarce for non-natural proteins. We also find that BBFlow is transferable to multimers – a class of proteins uncovered by current ensemble generation models.

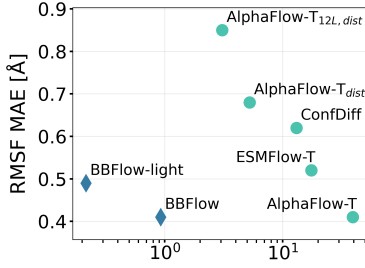

Figure 5: Trade-off between accuracy and speed of MD emulation. While other methods are either efficient or accurate, BBFlow performs well at both. The accuracy metric RMSF MAE and inference time are averaged over the ATLAS test set. More metrics can be found in Fig. A.3. BBFlow-light: App. A.6.

**Limitations**    As MD emulation model, BBFlow's scope is fundamentally limited to reproducing MD-derived distributions, thus, it cannot predict states that are very distant from equilibrium, such as alternative folding states, without being trained on correspondingly long MD trajectories. For predicting alternative states, specialized models, which are in turn less accurate at MD emulation, exist (see App. A.4). In terms of sampling transient contacts, BBFlow underperforms MSA-based approaches like AlphaFlow, indicating that evolutionary information is especially helpful for predicting rare events. Note that the requirement of an initial structure as input is not a limitation for MD emulators in practice and can be overcome (see App. A.1, Tab. A.1). Similar to the other backbone generation models in Sec. 1.1, BBFlow does not sample sidechain conformational ensembles or protein-ligand interactions, which is subject of further work.

## 5    Conclusion

The generation of MD-derived ensembles of proteins is a key task in many protein-related fields. Inspired by generative models for protein design, we propose BBFlow, a method for emulating MD by sampling ensembles with state-of-the-art performance in the trade-off between accuracy and efficiency. At the same time, BBFlow also avoids problems with *de novo* proteins and over-stabilization observed in current state-of-the-art models. Crucially, BBFlow is also applicable to multi-chain proteins, which are not covered by other ensemble generation models. We achieve this by introducing a conditional prior distribution and a geometric encoding of the protein's equilibrium structure as condition in a flow-matching model that is based on backbone geometry. This eliminates the need for evolutionary information and enables to train the model from scratch, without requiring weights from a folding model that is trained on a much larger dataset. We expect that both of these ideas – using a conditional prior in flow-matching and replacing evolutionary information by structure-conditioning – can also be applied to other problems in generative modeling and structural biology.

AlphaFlow [16] is widely used by practitioners to replace expensive MD simulations of proteins. We also see BBFlow as highly relevant in practice, given its significantly increased efficiency and transferability to multimers, allowing accurate MD emulation on a much larger scale. In particular, BBFlow can be applied in pipelines for *de novo* protein design, where it could enable the screening of generated structures for desired dynamics – a property that is challenging to incorporate into designed proteins so far.

**Acknowledgments**    This study received funding from the Klaus Tschira Stiftung gGmbH (HITS Lab). We also acknowledge support by the state of Baden-Württemberg through bwHPC and the German Research Foundation (DFG) through grant INST 35/1597-1 FUGG.

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

# A   Appendix

## A.1   On the requirement of an initial structure for MD emulation

Since BBFlow is an MD emulator, it also inherits the dependency on the initial structure from MD. AlphaFlow (without templates) can be applied as a sequence-to-structure model by relying on evolutionary information encoded in form of a Multiple Sequence Alignment (MSA). Since MD requires an initial structure, it can be assumed that an initial structure is available in all practical workflows that currently include expensive MD simulation. However, in a scenario where a sequence, but not the structure is available, BBFlow can be combined with a folding model and used as sequence-to-structure model that remains accurate and fast.

We show that the combination of AlphaFold2 and BBFlow remains both more accurate and more time-efficient at sequence-to-structure prediction than AlphaFlow's sequence-to-structure model at almost all metrics considered (Tab. A.1). For *de novo* proteins, the equilibrium structure that is passed to BBFlow and AlphaFlow-T was obtained by the folding model EMSFold during creation of the dataset (Sec. A.10), thus we already evaluate the combination of ESMFold and BBFlow, which acts as sequence-to-structure pipeline. BBFlow outperforms the sequence-to-structure model from AlphaFlow (without templates) in this setting (Tab. 2). Note that the inference time of the pipeline of AlphaFold 2 and BBFlow is still around 30 times faster than that of AlphaFlow because AlphaFlow requires a structure prediction for every conformation at every time step while the combination of AlphaFold 2 and BBFlow only requires a single structure prediction at the beginning.

We further investigate BBFlow's sensitivity with respect to good input equilibrium structures by evaluating its performance when passing distorted equilibrium structures (Tab. A.2) to the model. To this end, we add Gaussian noise to the Euclidean backbone coordinates of the ATLAS test set proteins and run inference with BBFlow. We find that the performance remains strong and decreases only slightly.

Table A.1: Performance of BBFlow as sequence-to-structure model. We evaluate the pipeline of predicting a structure from a sequence with AlphaFold 2 and passing this structure as input to BBFlow, compared to the direct sequence-to-structure model AlphaFlow. We report metrics as in Tab. 1.

|  | RMSF | | | Pw-RMSD | | | |
|---|---|---|---|---|---|---|---|
|  | $r$ ($\uparrow$) | MAE ($\downarrow$) | Median (1.48) | MAE ($\downarrow$) | DCCM $r$ ($\uparrow$) | PCA $\mathcal{W}_2$ ($\downarrow$) | Time [s] |
| AlphaFlow | 0.86 | 0.59 (0.01) | 1.51 | 1.35 (0.01) | **0.86** | **1.47** (0.03) | 32.0 |
| AlphaFold 2 + BBFlow | **0.87** | **0.52** (0.01) | **1.49** | **1.07** (0.01) | 0.85 | **1.47** (0.03) | **0.3+0.8** |

Table A.2: Performance of BBFlow with distorted equilibrium structures as input. We add Gaussian noise with the respective standard deviation to the backbone atoms equilibrium structure, or choose random MD conformations as equilibrium structure, and evaluate each model on the ATLAS test set. Units and settings as in Tab. 1.

|  | RMSF | | | Pw-RMSD | DCCM | PCA |
|---|---|---|---|---|---|---|
|  | $r$ ($\uparrow$) | MAE ($\downarrow$) | Median (MD=1.48) | MAE ($\downarrow$) | $r$ ($\uparrow$) | $\mathcal{W}_2$ ($\downarrow$) |
| BBFlow | 0.90 | 0.42 | 1.49 | 0.77 | 0.87 | 1.33 |
| BBFlow-0.1Å | 0.90 | 0.48 | 1.53 | 0.77 | 0.87 | 1.33 |
| BBFlow-0.2Å | 0.90 | 0.48 | 1.62 | 0.79 | 0.87 | 1.32 |
| BBFlow-random conf. | 0.90 | 0.47 | 1.56 | 0.83 | 0.85 | 1.56 |

## A.2   The role of nanosecond-timescale MD simulations of proteins

In the biophysics community, Molecular Dynamics (MD) is one of the most established method to study protein function and folding by analyzing conformational landscapes [20]. Most commonly, functional protein movements leading to molecular recognition, ligand binding, catalysis and protein folding span the timescale between microseconds to seconds [23, 48]. Sampling rare, large-scale conformational transitions with MD on the biologically relevant timescale, however, is often computationally prohibitive. Short-timescale simulations, while typically insufficient to observe rare

events, nonetheless provide valuable insight into local dynamics and mechanisms relevant to protein function.

Among a myriad of applications, short-time scale MDs in the range of hundreds of nanoseconds, as emulated by BBFlow and AlphaFlow, were employed to develop antibodies against NMDA receptors [40], to study allosteric regulation in enzymes [50, 37, 6, 6], to optimize biocatalysts for thermostability [49], and to shed light on the inhibition of SARS-CoV-2 NSP 13 helicase [35]. Given that BBFlow is capable of accurately emulating MD ensembles of 300 ns while being orders of magnitudes faster than current baselines, we believe it can become a useful tool in the hands of practitioners.

### A.3   Metrics for conformational ensembles

**RMSF**   The Root Mean Square Fluctuation (RMSF) of C$\alpha$ atoms measures the magnitude of positional deviations of individual residues across the set of conformations. For a given residue, these fluctuations are calculated in a reference frame that is defined by aligning the entire protein to the equilibrium structure and thus implicitly depend on the positions of all other residues. Consequently, RMSF can be interpreted as measure for flexibility, but also encodes global collective behaviour. As in AlphaFlow, we calculate the Pearson correlation between RMSF profiles (for an example see Fig. 2B) obtained from MD and generated ensembles in order to quantify how well the shapes of the profiles match. We also include the Mean Absolute Error (MAE) of per-residue RMSF to measure how well RMSF amplitudes are reproduced, and compare the median over all residues with the ground truth in order to quantify systematic over- or under-stabilization.

**Pairwise RMSD**   For each protein, we calculate the average C$\alpha$ RMSD between any two conformations $x$ as

$$\text{pwRMSD} \equiv \frac{1}{N^2} \sum_{i,j=1}^{N_{\text{confs}}} \text{RMSD}\left(x_i, x_j\right). \tag{13}$$

This quantifies the magnitude of conformational changes without requiring a specified reference state. We report the MAE of pairwise RMSD across all proteins.

**PCA**   A metric that explicitly accounts for conformational changes, and quantifies how well the respective conformations are captured, relies on the Principal Component Analysis of the C$\alpha$ positions across the sampled conformations. We project the generated states on the first two principal components obtained from MD, thus receiving a two-dimensional PCA-projection of each conformation. We report the Wasserstein-2-distance between the distributions of PCA-projections.

**DCCM**   Another metric that accounts for directional and long-range degrees of freedom is the Dynamic Cross Correlation Matrix (DCCM). It measures for each pair of residues $i, j$ whether they rather move in parallel, antiparallel or with uncorrelated relative direction. The entries are thus defined as

$$\text{DCCM}_{ij} \equiv \frac{\left\langle \left(\vec{x}_i - \langle\vec{x}_i\rangle\right) \cdot \left(\vec{x}_j - \langle\vec{x}_j\rangle\right)\right\rangle}{\sqrt{\left\langle \left(\vec{x}_i - \langle\vec{x}_i\rangle\right)^2 \right\rangle}\sqrt{\left\langle \left(\vec{x}_j - \langle\vec{x}_j\rangle\right)^2 \right\rangle}}, \tag{14}$$

where $\langle\ldots\rangle$ denotes the ensemble average and $\vec{x}_i$ denotes the C$\alpha$ atom position of residue $i$. For comparing the similarity of the DCCM matrices obtained with MD and with the generated ensemble, we report the Pearson correlation $r$ between all entries of both matrices.

#### A.3.1   Other metrics for protein ensembles

While the above metrics are frequently used and well established in the field of protein dynamics, Jing et al. [16] introduce two new, less established metrics for conformational ensembles, which we report for completeness in exhaustive evaluation tables in App. A.13.

**RMWD**   The root mean Wasserstein distance (RMWD) introduced in [16] assumes that atom positions in an ensemble are distributed according to three dimensional Gaussian distributions. It reports the Wasserstein distance between Gaussian distributions fitted to both the generated and the MD ensemble. However, the assumption that the atom positions are distributed according to

3D-Gaussians is a strong approximation – for comparing real ensembles, the strong correlation between individual atoms should be taken into account [28].

**Weak contacts $J$** Jing et al. [16] also report weak contacts as a metric defined as $C_\alpha$ pairs that are in contact (or not in contact) in the crystal structure but dissociate (or associate) in more than 10% of the ensemble structures, using an 8 Å distance cutoff. This metric is informative for larger conformational changes, however, the cutoff values of 8 Å and 10% are arbitrary and might not apply to more heterogeneous systems.

Table A.3: Evaluation of other ensemble generation methods (see App. A.4) as MD emulators on the ATLAS test set. Settings are the same as in Tab. 1. BBFlow-light is described in App. A.6.

| | RMSF | | | Pw-RMSD | DCCM | PCA | $J_{tr}$ | Time |
|---|---|---|---|---|---|---|---|---|
| | $r$ ($\uparrow$) | MAE ($\downarrow$) | Median (MD=1.48) | MAE ($\downarrow$) | $r$ ($\uparrow$) | $\mathcal{W}_2$ ($\downarrow$) | % ($\uparrow$) | [s] ($\downarrow$) |
| MDGen | 0.72 | 0.81 (0.01) | 0.62 | 2.05 (0.01) | 0.54 | 1.86 (0.03) | 27 | 0.15 |
| Str2Str | 0.52 | 7.80 (0.01) | 10.98 | 9.36 (0.01) | 0.52 | 1.63 (0.03) | 12 | 10.5 |
| ESMDiff | 0.69 | 1.57 (0.01) | 3.00 | 4.95 (0.01) | 0.75 | 1.84 (0.03) | 27 | 0.39 |
| BioEmu | 0.83 | 1.29 (0.01) | 2.34 | 2.84 (0.01) | 0.80 | 1.65 (0.04) | **36** | 1.9 |
| BBFlow-light | 0.89 | 0.48 (0.01) | 1.38 | 0.86 (0.01) | 0.86 | **1.32** (0.03) | 31 | **0.14** |
| BBFlow | **0.90** | **0.42** (0.01) | **1.49** | **0.77** (0.01) | **0.87** | 1.33 (0.03) | 29 | 0.77 |

## A.4 Other ensemble generation models

For sampling alternative folding states and general ensembles that must not strictly follow the distribution of states obtain via MD simulation with fixed runtime and temperature, several models have been developed recently. Lewis et al. [24] propose the generative model Bio-Emu, which is trained on a large custom dataset containing MDs of greatly varying lengths with an architecture similar to AlphaFold 2. A diffusion module is used for generating protein structures from the learned sequence representation, which relies on MSA. Another model trained on non-standardized MDs and also NMR data is ESMDiff [32], which relies on a Structure Language Model (SLM). In the proposed SLM, a discrete variational autoencoder is used to encode structure into tokens, whose relationship to the protein sequence is modeled by a language model. By fine-tuning the SLM, protein conformations can be sampled from the sequence. It has also been proposed to generate ensembles using models that are only trained on static structures, such as Str2Str [31], where noise is added to the equilibrium structure and a diffusion model is used to generate ensembles by partial denoising. Another example for such an approach is MSA-subsampling [47], where AlphaFold 2 is applied with modified MSA in order to sample alternative folding states.

Since one would not expect such models to sample the distribution induced by standardized MD to high accuracy, we only benchmark them for illustrative purposes in the appendix (Tab. A.3) and focus on the more relevant baselines AlphaFlow and ConfDiff in the main part of this paper.

Also for ATLAS-like ensembles, there is related work alongside ConfDiff; Li et al. [25] propose a modification of AlphaFlow, in which intermediate features of previous timesteps are re-used, leading to improved efficiency. However, neither code or model weights are published and only a limited subset of metrics is reported in the paper, prohibiting a direct comparison with BBFlow and other baselines.

## A.5 Ensemble generation for multi-chain proteins

**Multimeric systems** In order to assess BBFlow's performance for multimers, we simulate five multi-chain systems: a dimeric barnase-barstar (PDB: 1BGS), small homotrimer foldon (PDB: 1RFO), homodimeric fructokinase in its apo state (PDB: 5EY7), heterodimeric nanobody-SARS-CoV2 RBD complex (PDB: 7KGK) and heterodimeric nanobody-TNFRSF17 complex (PDB: 8HXR). We use simulation settings analogous to ATLAS (App. A.10). Selected complexes span total residue count between 80 and 590. We report BBFlow's performance in Tab. A.4.

**Baselines for multi-chain proteins** None of the baselines discussed in Sec. 1.1 or App. A.4 have been applied to multi-chain proteins before. Nonetheless, we tested the most relevant baseline, AlphaFlow [16], for the multi-chain systems. By default, AlphaFlow does not allow to parse multi-chain proteins. We thus modified the parser to accept multi-chain inputs.

More specifically, we took as an example a barnase-barstar complex (PDB: 1BGS), computed an MSA (pair mode: `unpaired_paired`, msa mode: `mmseqs2_uniref_env`) by querying the ColabFold mmseqs2 search server and used ColabFold's multimer parsing functions (`unserialize_msa(·)` followed by `generate_input_feature(·)`) to generate multimeric features that are compatible with AlphaFold's inference logics. We ran AlphaFlow inference with 10 timesteps by passing multimeric features as input and observed that the model fails to sample physical, protein-alike states despite good sequence coverage, as shown in Fig. A.1.

Table A.4: BBFlow's performance for the five multi-chain proteins described in App. A.5, evaluated in the settings of Tab. 1.

| | | RMSF | | Pw-RMSD | DCCM | PCA |
|---|---|---|---|---|---|---|
| | $r$ ($\uparrow$) | MAE ($\downarrow$) | Med. (MD=1.2) | MAE ($\downarrow$) | $r$ ($\uparrow$) | $\mathcal{W}_2$ ($\downarrow$) |
| BBFlow | 0.82 | 0.31 (0.01) | 1.40 | 0.41 (0.02) | 0.85 | 0.71 (0.06) |

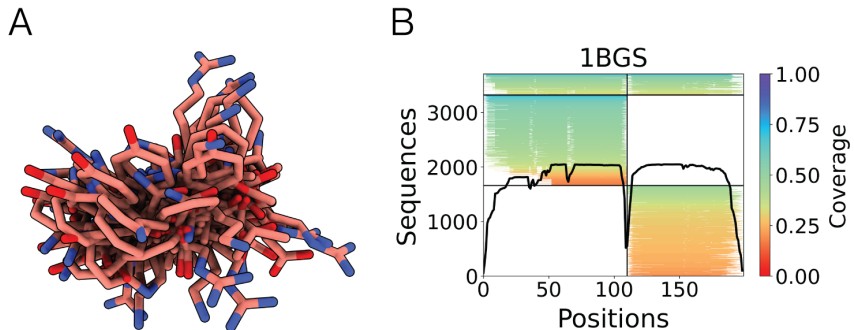

Figure A.1: AlphaFlow predicts unphysical states for the multi-chain barnase-barstar complex (PDB: 1BGS) if its parser is modified to accept multi-chain proteins. In (**A**), we show an example state sampled by AlphaFlow. (**B**) The parsed multimeric MSA features of the barnase-barstar complex show broad sequence coverage with many sequence hits for each of the chains.

### A.6 BBFlow-light model

We also train a BBFlow model with a set of hyperparameters different from those described in Sec. 4, chosen to increase efficiency on the cost of accuracy. We call the model BBFlow-light and report its performance in Tab. A.3. We find that it is around 200 times faster than AlphaFlow and around 10 times faster than AlphaFlow-T$_{12L, dist}$ while outperforming the distilled AlphaFlow models in most accuracy metrics.

In contrast to the vanilla BBFlow, BBFlow-light consists of 3 CFA message passing blocks instead of 6 and reduced node- and edge-feature dimensions of 96 and 48, respectively. BBFlow has around 18.2 M learnable parameters, while BBFlow-light contains only 2.5 M.

### A.7 Training algorithm

In Algorithm 1, we summarize the training procedure described in Sec. 3.

### A.8 Role of the hyperparameter $\xi$

We study the effect of the hyperparameter $\xi$ that controls the interpolation strength between the equilibrium structure and the unconditional prior in Eq. 8. Smaller $\xi$ values correspond to noisier

**Algorithm 1** Training of BBFlow
***

**Require:** Dataset $\mathcal{D} = \{(x_{\text{eq}}, \{x_1^{(i)}\}_{i=1}^M)\}$, with one equilibrium structure $x_{\text{eq}} \in \text{SE}(3)^N$, and several conformations $x_1^{(i)} \in \text{SE}(3)^N$ per protein
**Require:** Model $\hat{x}_\theta$
1:  **while** training **do**
2:    $(x_{\text{eq}}, x_1) \sim \mathcal{D}$                      ▷ Sample equilibrium structure and one conformation
3:    $x_0 \sim p_0(\cdot \mid x_{\text{eq}})$                    ▷ Sample noise from conditional prior (Eq. 8)
4:    $t \sim \mathcal{U}(0, 1)$                                ▷ Sample flow matching time
5:    $x_t = \gamma(x_0, x_1, t)$          ▷ Interpolated state $x_t \in \text{SE}(3)^N$, $\gamma$: geodesic between $x_0$ and $x_1$
6:    $v = v_{\text{SE}(3)}(x_t, t|x_1)$        ▷ Calculate ground-truth flow vector $v \in T_{x_t}\text{SE}(3)^N$ from Eq. 3
7:    $\hat{x}_1 = \hat{x}_\theta(x_t, t, x_{\text{eq}})$             ▷ Calculate model output $\hat{x}_1 \in \text{SE}(3)^N$
8:    $\hat{v} = v_{\text{SE}(3)}(x_t, t|\hat{x}_1)$          ▷ Calculate predicted flow vector $\hat{v} \in T_{x_t}\text{SE}(3)^N$ from Eq. 3
9:    $\mathcal{L}_{\text{FM}} = \|v - \hat{v}\|_{\text{SE}(3)}^2 + \mathcal{L}_{\text{aux}}(x_1, \hat{x}_1)$                ▷ Flow matching loss
10:   Update parameters using $\nabla_\theta \mathcal{L}_{\text{FM}}$
11: **end while**
***

initial states $x_0$, increasing ensemble diversity but also making training more challenging. Conversely, larger $\xi$ values bring $x_0$ closer to the equilibrium structure, facilitating convergence but potentially reducing ensemble diversity.

**Training-time ablation.** To quantify this tradeoff, we trained separate BBFlow models using different values of $\xi$. As shown in Tab. A.5, models trained with smaller $\xi$ produce more diverse ensembles (higher RMSF), while those with larger $\xi$ converge faster but exhibit reduced structural variability. We found that $\xi = 0.2$ provides a good balance between accuracy and diversity, and use this value in all reported experiments.

Table A.5: Ablation of models trained with different values of the hyperparameter $\xi$. Smaller $\xi$ increases ensemble diversity but slows convergence. Units and settings as in Tab. 1.

|  | RMSF $r$ ($\uparrow$) | RMSF MAE ($\downarrow$) | RMSF (MD=1.48) | Pw-RMSD MAE ($\downarrow$) | DCCM $r$ ($\uparrow$) | PCA $\mathcal{W}_2$ ($\downarrow$) |
|---|---|---|---|---|---|---|
| $\xi = 0.1$ | 0.88 | 0.54 | 1.53 | 0.87 | 0.86 | 1.35 |
| $\xi = 0.4$ | 0.89 | 0.44 | 1.39 | 0.89 | 0.85 | 1.33 |

**Inference-time ablation.** We further evaluated the model trained with $\xi = 0.2$ using different $\xi$ values during inference. As seen in Tab. A.6, reducing $\xi$ increases diversity (larger RMSF and PCA $\mathcal{W}_2$) but reduces agreement with molecular dynamics reference data. Increasing $\xi$ has the opposite effect, leading to more constrained ensembles.

Table A.6: Inference-time ablation of $\xi$ using the model trained with $\xi = 0.2$. Smaller $\xi$ increases ensemble diversity but reduces agreement with MD reference structures. Units and settings as in Tab. 1.

| $\xi$ | RMSF $r$ ($\uparrow$) | RMSF MAE ($\downarrow$) | RMSF (MD=1.48) | Pw-RMSD MAE ($\downarrow$) | DCCM $r$ ($\uparrow$) | PCA $\mathcal{W}_2$ ($\downarrow$) |
|---|---|---|---|---|---|---|
| 0.01 | 0.70 | 7.61 | 11.16 | 10.83 | 0.73 | 2.85 |
| 0.05 | 0.81 | 3.89 | 5.43 | 5.38 | 0.78 | 2.06 |
| 0.1 | 0.88 | 1.32 | 2.62 | 2.21 | 0.84 | 1.51 |
| 0.2 | 0.90 | 0.42 | 1.49 | 0.77 | 0.87 | 1.33 |
| 0.3 | 0.89 | 0.47 | 1.37 | 1.02 | 0.86 | 1.69 |
| 0.4 | 0.86 | 0.64 | 1.65 | 1.31 | 0.80 | 2.12 |
| 0.5 | 0.79 | 0.80 | 1.82 | 1.50 | 0.74 | 3.00 |
| 0.6 | 0.73 | 0.82 | 1.70 | 1.65 | 0.70 | 4.51 |

## A.9 Backbone dihedral distributions

In addition to the metrics above, we investigate the distribution of backbone dihedral angles across ensembles, commonly visualized in the field by Ramachandran plots. We show the Ramachandran

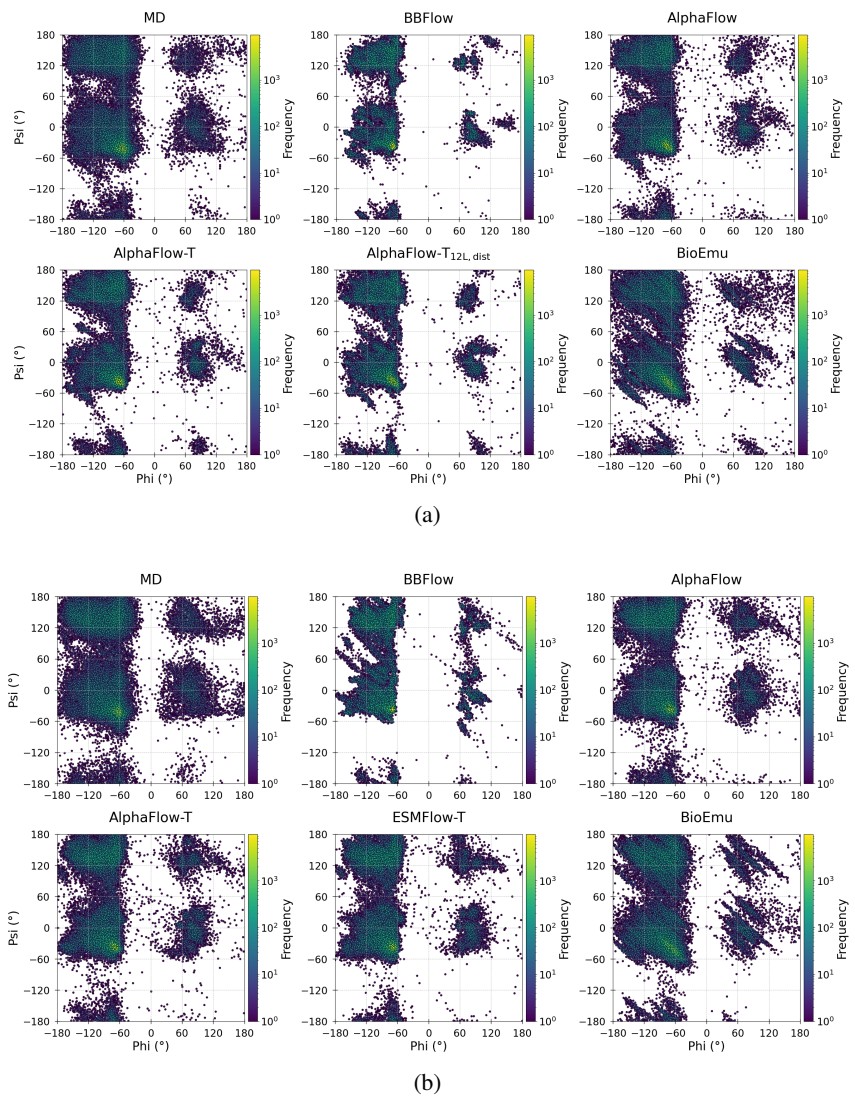

(a)

(b)

Figure A.2: Ramachandran plot, i.e. histogram of backbone dihedrals, for MD and generated ensembles of the representative proteins 6xrxA (a) and 6nl2A (b) from the ATLAS test set. We report Wasserstein distances of the distributions for the entire ATLAS test set in Tab. A.7.

plot of generated ensembles for a representative protein in Fig. A.2 and report the median of the Wasserstein distance to the MD distribution across the ATLAS test set in Tab. A.7. We find that BBFlow samples dihedral angles that deviate slightly more from MD than ESMFlow-T and AlphaFlow-T, but is competitive with the distilled models and those without templates.

Table A.7: Similarity of Ramachandran dihedral distribution across generated ensembles. We calculate the Wasserstein-2 distance between the dihedral angle distribution induced by MD and the respective generated ensemble and report medians across the ATLAS test set.

| Metric | AlphaFlow | BioEmu | AlphaFlow-T | EsmFlow-T | AlphaFlow-T$_D$ | AlphaFlow-T$_{12L,D}$ | BBFlow |
|---|---|---|---|---|---|---|---|
| Rama. $\mathcal{W}_2$ ($\downarrow$) | 0.53 | 0.66 | 0.48 | **0.47** | 0.51 | 0.55 | 0.52 |

## A.10  De novo proteins dataset

**Protein generation**  As described in Sec. 4.2, we assess the performance of BBFlow on a set of *de novo* proteins. We sample 20 protein backbones with FrameFlow [53] and RFdiffusion [46] for each length $L \in [60, 65, \ldots, 512]$. For each individual generated backbone, we carry out a self-consistency evaluation pipeline as previously proposed [52, 26] by designing 8 sequence with ProteinMPNN [10] and refolding candidate sequences with ESMfold [27]. We then compute the length distribution of the ATLAS dataset and select 50 refolded backbones that have a self-consistency RMSD (scRMSD) of $\leq 2.0$ Å to the originally generated backbone that follow a size distribution similar to the ATLAS dataset [42] each for FrameFlow and RFdiffusion.

**MD setup**  MD simulations are performed using GROMACS v2023 [1], utilizing the CHARMM27 all-atom force field. Proteins are embedded in a periodic dodecahedron box, ensuring a minimum separation of 1 nm from the box boundaries. The simulation system is hydrated using the TIP3P water model [18], and the ionic strength is adjusted to a NaCl concentration of 150 mM. An initial energy minimization is carried out for 5000 steps. The system undergoes NVT equilibration for 1 ns with a timestep of 2 fs, employing the leap-frog integrator. Temperature control is achieved at 300K using the Berendsen thermostat. This is followed by NPT equilibration for 1 ns, where the pressure is maintained at 1 bar using the Parrinello-Rahman barostat. The production run of the simulation extends over three 100 ns replicas. Throughout the simulations, covalent bonds involving hydrogen are constrained using the LINCS algorithm [13]. Long-range electrostatic interactions are treated using the Particle-Mesh Ewald (PME) method.

## A.11  ConfDiff inference setup

We evaluate ConfDiff using the ConfDiff-OF-r3-MD model, which is fine-tuned on the ATLAS dataset, available on GitHub[3]. We use the default hyperparameters for generating conformations.

## A.12  Additional figures

As extension of Fig. 5, we show the tradeoff between accuracy and speed with more accuracy metrics in Fig. A.3.

The construction of the conditional prior is visualized in Fig. A.4.

We show an extension of Fig. 2 to all metrics from Tab. 1 in Fig. A.6.

In Fig. A.5, we show the RMSF profiles of the multimers displayed in Fig. 4.

In Tab. A.8, we report the inference time of AlphaFlow's backbone and sidechain module, compared with BBFlow.

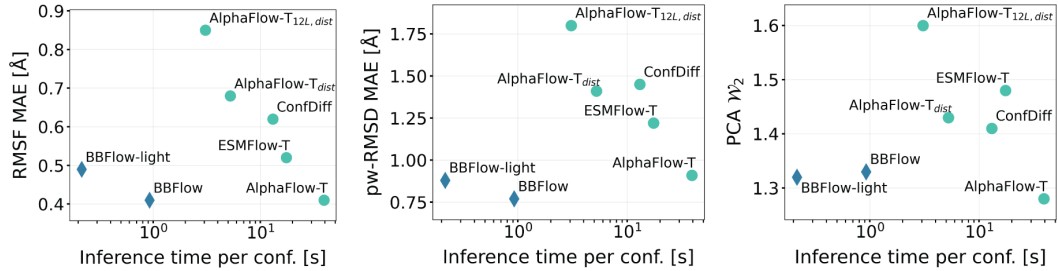

Figure A.3: Trade-off between accuracy and speed of MD emulation. While other methods are either efficient or accurate, BBFlow performs well at both. As extension of Fig.5, we show the accuracy metrics RMSF MAE, pairwise RMSD MAE and PCA $\mathcal{W}_2$ (all favorable if smaller). Both the accuracy metrics and inference time are averaged over the ATLAS test set. The other metrics (Pearson correlations and Median pw-RMSD) do not show such clear trends in terms of correlation with inference time and can be found in Tab. 1.

---

[3]`https://github.com/bytedance/ConfDiff`, commit 9cfae1c14121e423d8d455d03506c7e8ee580e48

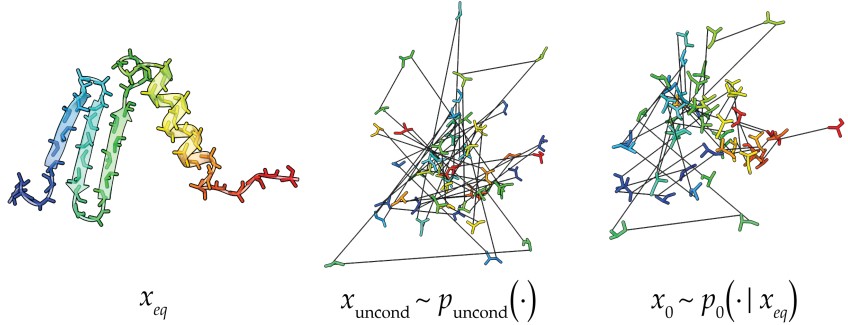

$$x_{eq} \qquad\qquad x_{\text{uncond}} \sim p_{\text{uncond}}(\cdot) \qquad\qquad x_0 \sim p_0(\cdot \,|\, x_{eq})$$

Figure A.4: Construction of the conditional prior (Eq. 8). For a given equilibrium structure as condition $x_{\text{eq}}$, we sample noise $x_{\text{uncond}}$ from the unconditional prior $p_{\text{uncond}}$ and interpolate along the geodesic between $x_{\text{uncond}}$ and $x_{\text{eq}}$ (Eq. 9) to obtain a sample $x_0$ from the proposed conditional prior $p_0(\cdot|x_{\text{eq}})$. In the experiments, we choose the hyperparameter $\xi = 0.2$; in the figure, we show a state sampled with $\xi = 0.5$ for better visualization.

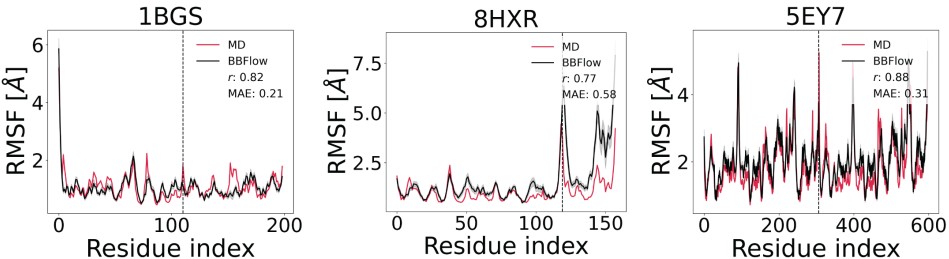

Figure A.5: RMSF profiles of three dimeric proteins whose DCCM matrices are depicted in Fig. 4 computed either with MD or BBFlow. Chain boundaries are indicated by black vertical lines.

Table A.8: Inference time by module of a single AlphaFlow and BBFlow forward pass on a 300 residue protein. The sidechain module of AlphaFlow is entirely separate from the backbone module and only takes a fraction of the backbone module's runtime. Inference time is therefore dominated by the backbone prediction task, for which BBFlow achieves a speedup.

| Module | Backbone [s] ($\downarrow$) | Sidechain [s] ($\downarrow$) |
|---|---|---|
| AlphaFlow-T | 32.5 | 0.12 |
| BBFlow | **0.8** | – |

## A.13 Exhaustive evaluation tables

We report the performance of BBFlow and baselines including the new metrics introduced in [16] (see Sec. A.3) in Tab. A.9 and Tab. A.11.

## A.14 Societal impact

We consider the societal impact of this work as mostly positive since understanding protein dynamics is essential for the development of new drugs, therapies and even materials, which outweighs potential risks.

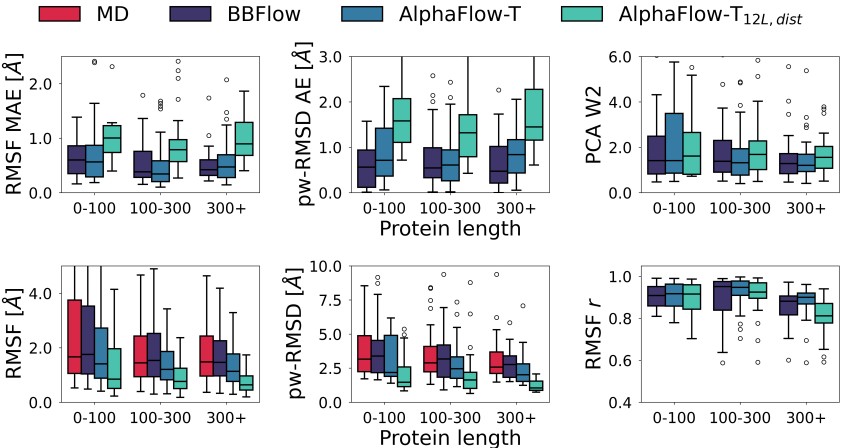

Figure A.6: Additional metrics for the performance of BBFlow, AlphaFlow-T and AlphaFlow-T$_{12L,dist}$ on the ATLAS test set for different protein lengths. We divide the protein lengths in three bins and calculate per-residue RMSF, RMSF MAE, RMSF correlation $r$, per-protein RMSD, the absolute error of pairwise RMSD and PCA $\mathcal{W}_2$ of each protein with length in the respective bin. The boxes depict minimum, maximum, median, and the 0.25 and 0.75 quantile.

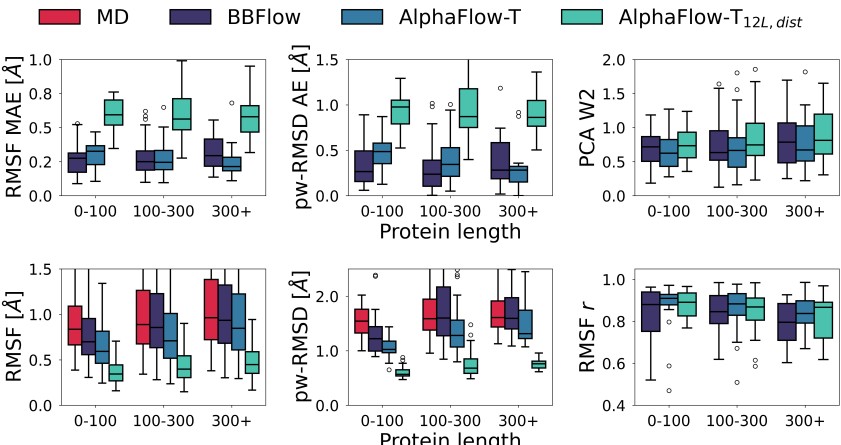

Figure A.7: Performance of BBFlow, AlphaFlow-T and AlphaFlow-T$_{12L,dist}$ on the *de novo* protein dataset for different protein lengths. We divide the protein lengths in three bins and calculate per-residue RMSF, RMSF MAE, RMSF correlation $r$, per-protein RMSD, the absolute error of pairwise RMSD and PCA $\mathcal{W}_2$ of each protein with length in the respective bin. The boxes depict minimum, maximum, median, and the 0.25 and 0.75 quantile.

Table A.9: Evaluation on the ATLAS dataset including the metrics introduced in [16] (see Sec. A.3). RMSF and RMWD are calculated from C$\alpha$ atoms.

| | AlphaFlow | BioEmu | ConfDiff | AlphaFlow-T | EsmFlow-T | AlphaFlow-T$_{12L,dist}$ | BBFlow |
|---|---|---|---|---|---|---|---|
| Pairwise RMSD (=2.9) | 2.89 | 4.29 | 3.43 | 2.18 | 2.0 | 1.4 | 3.09 |
| Pairwise RMSD $r$ | 0.48 | 0.23 | 0.62 | 0.94 | 0.85 | 0.76 | 0.82 |
| Pairwise RMSD MAE | 1.35 | 2.84 | 1.45 | 0.91 | 1.22 | 1.8 | 0.77 |
| C$_\alpha$ RMSF (=1.48) | 1.51 | 2.34 | 2.0 | 1.17 | 0.94 | 0.68 | 1.49 |
| Global RMSF $r$ | 0.58 | 0.46 | 0.7 | 0.91 | 0.84 | 0.74 | 0.84 |
| Per target RMSF $r$ | 0.86 | 0.83 | 0.88 | 0.92 | 0.92 | 0.9 | 0.9 |
| Per target RMSF MAE | 0.59 | 1.29 | 0.62 | 0.41 | 0.52 | 0.85 | 0.42 |
| Global DCCM $r$ | 0.8 | 0.73 | 0.85 | 0.88 | 0.87 | 0.85 | 0.84 |
| Per target DCCM $r$ | 0.86 | 0.80 | 0.86 | 0.89 | 0.89 | 0.87 | 0.87 |
| Per target DCCM MAE | 0.15 | 0.19 | 0.14 | 0.12 | 0.12 | 0.13 | 0.15 |
| Root mean $\mathcal{W}_2$-distance | 2.38 | 3.36 | 2.56 | 1.72 | 1.91 | 2.13 | 1.93 |
| – Translation contrib. | 2.17 | 2.51 | 2.02 | 1.47 | 1.52 | 1.73 | 1.65 |
| – Variance contrib. | 1.18 | 1.99 | 1.22 | 0.82 | 0.92 | 1.2 | 0.93 |
| MD PCA $\mathcal{W}_2$-distance | 1.47 | 1.65 | 1.41 | 1.28 | 1.48 | 1.6 | 1.33 |
| Joint PCA $\mathcal{W}_2$-distance | 2.26 | 2.90 | 2.19 | 1.58 | 1.76 | 1.93 | 1.72 |
| % PC-sim $> 0.5$ | 43.85 | 24.49 | 37.76 | 44.6 | 47.94 | 39.12 | 40.1 |
| Weak contacts $J$ | 0.62 | 0.46 | 0.63 | 0.62 | 0.59 | 0.56 | 0.57 |
| Transient contacts $J$ | 0.41 | 0.36 | 0.39 | 0.47 | 0.47 | 0.24 | 0.29 |
| Time [s] | 32.0 | 1.9 | 20.2 | 32.6 | 11.2 | 1.2 | 0.8 |

Table A.10: Evaluation on the *de novo* dataset including the metrics introduced in [16] (see Sec. A.3). RMSF and RMWD are calculated from C$\alpha$ atoms.

| | AlphaFlow | BioEmu | ConfDiff | AlphaFlow-T | EsmFlow-T | AlphaFlow-T$_{12L,dist}$ | BBFlow |
|---|---|---|---|---|---|---|---|
| Pairwise RMSD (=1.59) | 8.08 | 7.9 | 7.27 | 1.25 | 1.2 | 0.68 | 1.47 |
| Pairwise RMSD $r$ | 0.2 | 0.13 | 0.24 | 0.86 | 0.86 | 0.83 | 0.7 |
| Pairwise RMSD MAE | 7.4 | 8.29 | 7.26 | 0.38 | 0.43 | 0.97 | 0.32 |
| C$_\alpha$ RMSF (=0.91) | 7.09 | 7.56 | 6.35 | 0.74 | 0.68 | 0.41 | 0.87 |
| Global RMSF $r$ | 0.27 | 0.25 | 0.28 | 0.86 | 0.86 | 0.83 | 0.77 |
| Per target RMSF $r$ | 0.47 | 0.6 | 0.62 | 0.89 | 0.89 | 0.87 | 0.84 |
| Per target RMSF MAE | 4.76 | 4.24 | 3.82 | 0.25 | 0.28 | 0.58 | 0.26 |
| Global DCCM $r$ | 0.52 | 0.53 | 0.54 | 0.79 | 0.8 | 0.77 | 0.77 |
| Per target DCCM $r$ | 0.58 | 0.64 | 0.65 | 0.85 | 0.86 | 0.83 | 0.83 |
| Per target DCCM MAE | 0.22 | 0.21 | 0.21 | 0.11 | 0.1 | 0.11 | 0.14 |
| Root mean $\mathcal{W}_2$-distance | 10.17 | 7.32 | 8.39 | 1.02 | 0.98 | 1.27 | 1.2 |
| – Translation contrib. | 8.01 | 4.25 | 6.91 | 0.9 | 0.85 | 1.0 | 1.06 |
| – Variance contrib. | 5.5 | 5.2 | 4.29 | 0.45 | 0.47 | 0.73 | 0.54 |
| MD PCA $\mathcal{W}_2$-distance | 1.64 | 1.53 | 1.72 | 0.66 | 0.63 | 0.75 | 0.67 |
| Joint PCA $\mathcal{W}_2$-distance | 9.08 | 5.23 | 7.32 | 0.95 | 0.89 | 1.13 | 1.09 |
| % PC-sim $> 0.5$ | 9.86 | 7.46 | 16.09 | 48.05 | 50.19 | 40.99 | 39.37 |
| Weak contacts $J$ | 0.38 | 0.39 | 0.44 | 0.61 | 0.61 | 0.57 | 0.57 |
| Transient contacts $J$ | 0.17 | 0.23 | 0.15 | 0.55 | 0.56 | 0.38 | 0.32 |
| Time [s] | 32.0 | 1.9 | 20.2 | 32.6 | 11.2 | 1.2 | 0.8 |

Table A.11: Evaluation of ensemble generation methods from App. A.4 as MD emulators on the ATLAS dataset including the metrics introduced in [16] (see Sec. A.3). RMSF and RMWD are calculated from C$\alpha$ atoms.

| | MDGen | Str2Str | ESMDiff | BioEmu | BBFlow-light | BBFlow |
|---|---|---|---|---|---|---|
| Pairwise RMSD (=2.9) | 1.34 | 13.29 | 5.36 | 4.29 | 2.6 | 3.09 |
| Pairwise RMSD $r$ | 0.48 | 0.15 | 0.22 | 0.23 | 0.81 | 0.82 |
| Pairwise RMSD MAE | 2.05 | 9.36 | 4.95 | 2.84 | 0.86 | 0.77 |
| C$_\alpha$ RMSF (=1.48) | 0.62 | 10.98 | 3.0 | 2.34 | 1.38 | 1.49 |
| Global RMSF $r$ | 0.49 | 0.28 | 0.31 | 0.46 | 0.82 | 0.84 |
| Per target RMSF $r$ | 0.72 | 0.52 | 0.69 | 0.83 | 0.89 | 0.9 |
| Per target RMSF MAE | 0.81 | 7.8 | 1.57 | 1.29 | 0.48 | 0.42 |
| Global DCCM $r$ | 0.35 | 0.46 | 0.71 | 0.73 | 0.83 | 0.84 |
| Per target DCCM $r$ | 0.54 | 0.52 | 0.75 | 0.80 | 0.86 | 0.87 |
| Per target DCCM MAE | 0.23 | 0.26 | 0.19 | 0.19 | 0.16 | 0.15 |
| Root mean $\mathcal{W}_2$-distance | 2.59 | 9.58 | 4.84 | 3.36 | 2.04 | 1.93 |
| – Translation contrib. | 2.15 | 4.74 | 3.48 | 2.51 | 1.75 | 1.65 |
| – Variance contrib. | 1.32 | 8.11 | 2.63 | 1.99 | 0.96 | 0.93 |
| MD PCA $\mathcal{W}_2$-distance | 1.86 | 1.63 | 1.84 | 1.65 | 1.32 | 1.33 |
| Joint PCA $\mathcal{W}_2$-distance | 2.47 | 6.49 | 3.94 | 2.90 | 1.81 | 1.72 |
| % PC-sim > 0.5 | 13.74 | 2.5 | 21.1 | 24.49 | 37.43 | 40.1 |
| Weak contacts $J$ | 0.5 | 0.3 | 0.49 | 0.46 | 0.5 | 0.57 |
| Transient contacts $J$ | 0.27 | 0.12 | 0.36 | 0.36 | 0.31 | 0.29 |
| Time [s] | 0.15 | 10.5 | 0.39 | 1.9 | 0.14 | 0.77 |

Table A.12: Ablation study including the metrics introduced in [16]. Extension of Tab. 3.

| | BBFlow | a | b | c | d | e |
|---|---|---|---|---|---|---|
| Pairwise RMSD (=2.9) | 3.09 | 2.58 | 2.35 | 2.62 | 2.4 | 11.12 |
| Pairwise RMSD $r$ | 0.82 | 0.74 | 0.81 | 0.83 | 0.82 | 0.04 |
| Pairwise RMSD MAE | 0.77 | 1.15 | 0.9 | 0.82 | 0.93 | 7.08 |
| C$_\alpha$ RMSF (=1.48) | 1.49 | 1.55 | 1.3 | 1.25 | 1.34 | 8.26 |
| Global RMSF $r$ | 0.84 | 0.72 | 0.81 | 0.85 | 0.81 | 0.19 |
| Per target RMSF $r$ | 0.9 | 0.88 | 0.9 | 0.9 | 0.87 | 0.44 |
| Per target RMSF MAE | 0.42 | 0.52 | 0.48 | 0.42 | 0.54 | 5.88 |
| Global DCCM $r$ | 0.84 | 0.82 | 0.85 | 0.85 | 0.84 | 0.48 |
| Per target DCCM $r$ | 0.87 | 0.85 | 0.86 | 0.88 | 0.85 | 0.55 |
| Per target DCCM MAE | 0.15 | 0.16 | 0.16 | 0.14 | 0.16 | 0.23 |
| Root mean $\mathcal{W}_2$-distance | 1.93 | 2.09 | 1.95 | 1.97 | 2.19 | 7.81 |
| – Translation contrib. | 1.65 | 1.75 | 1.7 | 1.67 | 1.82 | 4.22 |
| – Variance contrib. | 0.93 | 1.04 | 0.98 | 0.94 | 1.03 | 6.3 |
| MD PCA $\mathcal{W}_2$-distance | 1.33 | 1.46 | 1.44 | 1.32 | 1.47 | 1.32 |
| Joint PCA $\mathcal{W}_2$-distance | 1.72 | 1.93 | 1.8 | 1.82 | 1.96 | 4.53 |
| % PC-sim > 0.5 | 40.1 | 37.39 | 46.77 | 41.82 | 38.4 | 9.2 |
| Weak contacts $J$ | 0.57 | 0.52 | 0.45 | 0.55 | 0.52 | 0.31 |
| Transient contacts $J$ | 0.29 | 0.3 | 0.32 | 0.31 | 0.26 | 0.1 |

