# OpenReview forum: "Learning conformational ensembles of proteins based on backbone geometry"
_NeurIPS.cc/2025/Conference — NeurIPS 2025 poster_

### Official Review · Reviewer_ndHw · 2025-06-28

**Clarity:** 3
**Significance:** 2
**Originality:** 2
**Rating:** 4
**Confidence:** 4

**Summary:**

The paper presents a SE(3)-flow matching method for emulating MD ensembles, given access to a seed structure. Specifically, the model learns a flow from a prior centered around the seed structure to the MD ensemble using relatively lightweight (i.e., pairformer-free) architectures. When trained on the ATLAS dataset, it is shown to perform comparable to AlphaFlow while possessing significantly faster runtimes. Despite being trained only on naturally occuring protein monomers, the model is also shown to have nontrivial performance on de novo proteins and for dimers.

**Questions:**

See above.

**Ethical Concerns:**

["NO or VERY MINOR ethics concerns only"]

**Final Justification:**

I maintain my positive assessment of the work and its empirical results.

**Limitations:**

Yes.

**Quality:**

3

**Strengths And Weaknesses:**

**Strengths**
* The paper's architectural contribution is conceptually reasonable and a noteworthy improvement in terms of efficiency upon prior methods.
* The paper's main claims are fully supported by the experiments and the paper is well written.
* The paper provides the first demonstration of an MD ensemble emulator for dimers and de novo, a noteworthy step forward.


**Weaknesses**
* The main challenge with ensemble emulators is that practitioners are interested in sampling rare events, such as transitions to kinetically separated Markov states. It is very hard to assess whether these events are captured using the provided metrics. The transient and weak contacts metrics from AlphaFlow come closest to measuring these phenomena, but the presented method lags behind AlphaFlow by those metrics. The authors have also deferred these metrics to the appendix, which weakens the paper.
* Why is BioEmu not included in the results in the main text?
* Please provide the full set of metrics for the de novo designed proteins.

---

> ### Author Rebuttal · Authors · 2025-07-30
>
> We thank the reviewer for their feedback and constructive critique. We are glad to read the reviewer finds our paper well-written, conceptually reasonable and the efficiency improvement noteworthy. Below we address weaknesses mentioned in the review.
>
>
> &nbsp;
>
> *A4.1 - Addressing W1: Rare events sampling*
>
>  We agree with the reviewer that sampling conformational changes triggered by rare events is interesting for practitioners. However, these typically occur on MD timescales well beyond 300ns, which is the MD timescale of the ATLAS dataset. As we state in section 1.1, A.3, 'limitations' and 'Further baselines' in the original submission, sampling such large conformational changes and alternative states is a task that is different from the one solved by AlphaFlow and BBFlow.
>
>  In AlphaFlow and BBFlow, MD is emulated under fixed, standardized settings.
>  The ground truth ensembles strongly depend on these settings (temperature, timescale, forcefield). Thus, evaluating distributional accuracy quantitatively (and not rather qualitatively as in BioEmu) is only sensible if these settings are fixed. This becomes apparent in the poor performance of BioEmu in all metrics on ATLAS as done in Table A.2 of the original submission and on de novo proteins in Table R16. This poor performance is expected because BioEmu does not claim to solve this specific task, conversely it can be expected that BBFlow and AlphaFlow would perform poorly if evaluated on BioEmu's task.
>
>  We would like to note that not only the prediction of rare events and far alternative states but also emulation of nanosecond-timescale MD has a lot of applications. These kinds of MD simulations are employed on a regular basis by practitioners for multiple different purposes. To state a few concrete examples: MD on the 300 ns timescale can be used for assessing opening-closing movements, hinge movements or flexibility of binding pockets and is useful for rational enzyme design. [1, 2, 3].
>
> ---
>
> [1] Raubenolt, Bryan A., et al. "Molecular dynamics simulations of the flexibility and inhibition of SARS-CoV-2 NSP 13 helicase." Journal of Molecular Graphics and Modelling 112 (2022)
>
> [2] Kiss, Gert et al. "Molecular dynamics simulations for the ranking, evaluation, and refinement of computationally designed proteins." Methods in Enzymology. Vol. 523. Academic Press, 2013. 145-170.
>
> [3] Wijma, Hein J. et al. "A computational library design protocol for rapid improvement of protein stability: FRESCO." Protein engineering: Methods and protocols. New York, NY: Springer New York, 2017. 69-85.
>
>
> &nbsp;
>
> *A4.2 - Addressing W1: Contact metrics in appendix*
>
> As we argue in section A.2.1, we focus on metrics that are established in the field of biomolecular dynamics and provide the contact metrics, which were only recently introduced in AlphaFlow, for completeness in the appendix. As stated in that section, the metrics contain arbitrary thresholds of 8 Å and 10\%. BBFlow indeed performs slightly worse than AlphaFlow but still better than the distilled model (Table A.4). We regard the distilled model as relevant baseline since it is the only AlphaFlow model that comes close to BBFlow in terms of efficiency, demonstrating BBFlow's favourable performance in the tradeoff between efficiency and accuracy. Notably, BBFlow also performs better in the contact metric than BioEmu (Table A.4), which is commonly considered as good model for off-equilibrium sampling. However, we included BBFlow's slightly worse performance in the contact metrics as limitation in the 'limitations' section in the main text.
>
> We would like to note that we ran an additional experiment during the rebuttal phase that shows that there is no clear trend, which model is generally better at sampling off-equilibrium states. In addition to the contact metrics, we introduce another metric that measures the difference between equlibrium structure and generated samples. Instead of using contacts as difference measure between states, we explicitly calculate the RMSD between samples and equilibrium structure for each protein and report the values in Table R13.
> We find the reversed trend - BBFlow being better than AlphaFlow-T at capturing off-equilibrium states.
> In terms of RMSD, BBFlow samples states that are further away from the equilibrium structure, while AlphaFlow-T samples closer to the equilibrium structure. We believe this demonstrates that slight performance advantages may depend on the metric definition and that there is no clear candidate that is generally better at sampling larger conformational changes.
>
> | Method       | Mean RMSD to Eq. Structure [Å] |
> |--------------|--------------------------|
> | MD           | 3.0                      |
> | BBFlow       | **2.5**                  |
> | AlphaFlow-T  | 2.1                      |
> | BioEMU       | 4.0                      |
> | AlphaFlow    | **3.5**                  |
>
> *Table R13. Deviation from the equilibrium structure on the ATLAS test set. Closer to MD is better.*
>
>
> &nbsp;
>
> *A4.3 - Addressing W2: Omission of BioEmu from the main text*
>
> As discussed in answer A4.1, we find reporting BioEmu’s performance on the ATLAS test set would place it in an unfavourable evaluation setting, as BioEmu is not designed to emulate 300ns of MD with distributional accuracy. It is rather designed to sample off-equilibrium states by training on longer MD timescales and fine-tuning on folding transitions with experimental data. We think this makes a comparison to BBFlow and other models trained on ATLAS dataset (300 ns) unfair (for BioEmu) and misleading and should not be emphasized in the main text. We stress this in Section 4 of the original submission under Further Baselines.
>
> Appendix Tables A.2 and A.5 of the original submission illustrate that BioEmu, indeed, performs poorly across nearly all metrics on the ATLAS test set in standardized settings. We also predicted ensembles for de novo proteins with BioEmu during the rebuttal period and report its performance in Table R14. Probably due to shallow evolutionary information for de novo sequences, BioEmu performs even worse for those proteins. Conversely, it can be expected that BBFlow and AlphaFlow would perform poorly if evaluated on BioEmu's task.
>
> |       | RMSF r | RMSF MAE | RMSF (MD=1.48) | pw-RMSD (MD=2.9) | pw-RMSD MAE | DCCM r | MD PCA W2 | Time/conf|
> |:------|-------:|---------:|---------------:|-----------------:|------------:|-------:|----------:|--:|
> | BioEmu |   0.6  |     4.24 |           7.56 |             7.9  |        8.29 |   0.64 |      1.53 | 1.9 |
> | BBFlow |   0.84 |     0.26 |           0.87 |             1.47 |        0.32 |   0.83 |      0.67 | 0.8 |
>
> *Table R14. Performance of BioEmu (and BBFlow) on the de novo proteins from Table 2 in the original submissions. Units and settings as in Table 2. Full set of metrics in Table R15.*
>
>
> &nbsp;
>
> *A4.4 - Addressing W3: Provide full set of metrics for de novo proteins*
>
> We thank the reviewer for their suggestion. We report the full set of metrics for de novo proteins in Table R15.
>
> | metric                  |    AF | ConfDiff |  AF-T |  EF-T | AF-T-D | AF-T-12L-D | BioEmu | BBFlow |
> |:------------------------|------:|---------:|------:| -----:|-------:|-----------:|-------:|-------:|
> | Pairwise RMSD (=2.9)    |  8.08 |     7.27 |  1.25 |  1.2  |   0.87 |       0.68 |   7.9  |   1.47 |
> | Pairwise RMSD r         |  0.2  |     0.24 |  0.86 |  0.86 |   0.82 |       0.83 |   0.13 |   0.7  |
> | Pairwise RMSD MAE       |  7.4  |     7.26 |  0.38 |  0.43 |   0.77 |       0.97 |   8.29 |   0.32 |
> | CA RMSF (=1.48)         |  7.09 |     6.35 |  0.74 |  0.68 |   0.51 |       0.41 |   7.56 |   0.87 |
> | Global RMSF r           |  0.27 |     0.28 |  0.86 |  0.86 |   0.84 |       0.83 |   0.25 |   0.77 |
> | Per target RMSF r       |  0.47 |     0.62 |  0.89 |  0.89 |   0.88 |       0.87 |   0.6  |   0.84 |
> | Per target RMSF MAE     |  4.76 |     3.82 |  0.25 |  0.28 |   0.46 |       0.58 |   4.24 |   0.25 |
> | Global DCCM r           |  0.52 |     0.54 |  0.79 |  0.8  |   0.79 |       0.77 |   0.53 |   0.77 |
> | Per target DCCM r       |  0.58 |     0.65 |  0.85 |  0.86 |   0.84 |       0.83 |   0.64 |   0.83 |
> | Per target DCCM MAE     |  0.22 |     0.21 |  0.11 |  0.1  |   0.11 |       0.11 |   0.21 |   0.14 |
> | Root mean W2-distance   | 10.17 |     8.39 |  1.02 |  0.98 |   1.1  |       1.27 |   7.32 |   1.2  |
> | -- Translation contrib. |  8.01 |     6.91 |  0.9  |  0.85 |   0.88 |       1    |   4.25 |   1.06 |
> | -- Variance contrib.    |  5.5  |     4.29 |  0.45 |  0.47 |   0.6  |       0.73 |   5.2  |   0.54 |
> | MD PCA W2-distance      |  1.64 |     1.72 |  0.66 |  0.63 |   0.69 |       0.75 |   1.53 |   0.67 |
> | Joint PCA W2-distance   |  9.08 |     7.32 |  0.95 |  0.89 |   0.97 |       1.13 |   5.23 |   1.09 |
> | % PC-sim > 0.5          |  9.86 |    16.09 | 48.05 | 50.19 |  48.25 |      40.99 |   7.46 |  39.37 |
> | Weak contacts J         |  0.38 |     0.44 |  0.61 |  0.61 |   0.54 |       0.57 |   0.39 |   0.57 |
> | Transient contacts J    |  0.17 |     0.15 |  0.55 |  0.56 |   0.51 |       0.38 |   0.23 |   0.32 |
>
> *Table R15.: Evaluation on the de novo proteins from Table 2 including the metrics introduced in AlphaFlow. Units and settings as in Table A.2 and Table 1, respectively.*

---

> > ### Comment · Reviewer_ndHw · 2025-08-06
> >
> > I appreciate the authors' response. As noted in my original review, I am willing to recommend acceptance on the basis of the model's efficiency and competitive performance with respect to AlphaFlow.
> >
> > However, it is important that the performance of the method be transparently conveyed. I request that the authors agree to include the contact-based metrics and comparisons with BioEmu in the revision of the main text, as a condition of acceptance.
> >
> > The contact-based metrics are a means of assessing the correctness of predicted conformational changes. As such, they are qualitatively different than the additional metric of RMSD to equilibrium structure suggested by the authors in rebuttal. I do not find the authors' suggestion that the metrics are arbitrary and the proposal of yet another metric, simply to argue that the relative performance of methods can change, to be convincing.

---

> > > ### Author Response · Authors · 2025-08-06
> > >
> > > We thank the reviewer for responding to our rebuttal and appreciate that the reviewer recommends to accept our paper.
> > >
> > > In the final version, we will indeed report the contact metrics and BioEmu's performance in the main text. Please note that our method outperforms BioEmu in all metrics, also the contact metrics. This can be expected because BioEmu is not trained to generate ATLAS-like ensembles (300ns, 300K), in contrast to other baselines like AlphaFlow. We will thus add a respective disclaimer in the main text and table caption.
> > >
> > > We are happy to answer any further questions if needed!

---

### Official Review · Reviewer_EssD · 2025-07-01

**Clarity:** 3
**Significance:** 2
**Originality:** 2
**Rating:** 4
**Confidence:** 4

**Summary:**

The paper presents BBFlow, a deep generative model designed to learn conformational ensembles of proteins based solely on backbone geometry. By reframing the problem as a conditional generation task conditioned on equilibrium structural information, the authors aim to eliminate reliance on evolutionary data such as MSAs and large pre-trained protein folding models. The key contributions include a conditional encoding scheme that uses only equilibrium backbone geometry and a novel conditional prior integrated into the flow matching process.

**Questions:**

1.	How is the hyperparameter $\xi$ in the conditional prior chosen? Could the authors provide insights into how sensitive model performance is to this parameter?

2.	In Table A.3, BBFlow performs significantly better on multi-chain systems than on the ATLAS test set (Table 1). Given the increased complexity and out-of-distribution nature of multi-chain systems, this result is unintuitive. Could the authors offer an explanation?

**Ethical Concerns:**

["NO or VERY MINOR ethics concerns only"]

**Final Justification:**

The rebuttal has resolved most of my concerns. Thus, I will raise the score.

**Limitations:**

The main limitations of the work are restricted empirical validation and marginal empirical gains over existing baselines like AlphaFlow-T.

**Paper Formatting Concerns:**

No formatting issues were identified in the current version of the manuscript.

**Quality:**

3

**Strengths And Weaknesses:**

Pros:

•	The motivation to replace evolutionary features with pure structural priors is well-articulated and technically compelling.

•	BBFlow demonstrates state-of-the-art performance in terms of the speed–accuracy trade-off on a widely accepted benchmark.

•	The model architecture—particularly the omission of residue indices—makes BBFlow naturally extendable to multi-chain systems, broadening its applicability.

Cons:

Despite the promising framework, several issues limit the practical impact and technical novelty of the work:

•	Given that BBFlow is conditioned on equilibrium structure, its most relevant baseline is AlphaFlow-T. The performance improvements over AlphaFlow-T are modest. In addition, as shown in the ablation study (Table 3), removing the conditional prior results in performance comparable to AlphaFlow-T, suggesting that the empirical impact of equilibrium structure encoding is limited.

•	Since BBFlow depends heavily on equilibrium structures for conditioning, the quality and accuracy of these structures could critically influence performance. The paper does not discuss robustness to noisy or approximate inputs, which would be important in real-world applications.

•	The model is primarily evaluated on short-timescale MD simulations, where conformational variability is relatively small. The paper would benefit from additional results on proteins with complex, multi-state dynamics, such as those in BioEmu [2], to demonstrate real-world utility.

•	There is no direct discussion or comparison with AlphaFlow-Lit [1], which also uses feature-conditioned generative modeling for efficient protein dynamics and serves as a relevant baseline.


[1] Li S, Li M, Wang Y, et al. Improving AlphaFlow for efficient protein ensembles generation[J]. arXiv preprint arXiv:2407.12053, 2024.

[2] Lewis, Sarah, et al. "Scalable emulation of protein equilibrium ensembles with generative deep learning." bioRxiv (2024): 2024-12.

---

> ### Author Rebuttal · Authors · 2025-07-30
>
> We thank the reviewer for the time they invested in reading the paper and their helpful feedback. We are happy that the reviewer finds our method technically compelling, sees it as state-of-the-art on a widely accepted benchmark and recognizes its applicability to multi-chain systems.
> For resolving the concerns raised in the review, we conducted additional experiments, which we will include in the final version of the paper. In the following, we address all points individually.
>
> &nbsp;
>
> *A3.1 - Addressing W1: Performance improvements over AlphaFlow-T are modest*
>
> We would like to note that performance is not only assessed solely via accuracy but also efficiency has to be considered. In this combination, the tradeoff between accuracy and efficiency, BBFlow's improvement over AlphaFlow-T is large: BBFlow is 40 times faster, at comparable accuracy. Additionally, it covers multi-chain systems, an important class of proteins for which AlphaFlow is not applicable. Both of these points are also recognized by other reviewers.
>
> &nbsp;
>
> *A3.2 - Addressing W1: The empirical impact of equilibrium structure encoding is limited*
>
> As the reviewer states, removing the conditional prior indeed reduces performance (Table 3, line b). But removing the equilibrium structure encoding and keeping the conditional prior (line e) reduces performance even stronger. Thus, both of these innovations are needed to achieve state-of-the-art performance. Of course, line b suggests that the conditional prior applied within AlphaFlow-T (i.e. without eq. structure encoding but with MSA and Evoformer) might lead to the same performance in terms of accuracy, however, efficiency would be greatly reduced. Thus, also the equilibrium structure encoding can be regarded as crucial, especially for efficiency.
>
> &nbsp;
>
> *A3.3 - Addressing W2: Robustness under noisy or bad quality equilibrium structures*
>
> **Short answer:**
>
> If the quality of the equilibrium structure is reduced, BBFlow's performance only decreases slightly, indicating robustness under noisy equilibrium structures.
>
> **Long answer:**
>
> We agree with the reviewer that robustness under changes of the equilibrium structure is a good consistency check and also relevant in practice. In Table A.1 of the original submission, we reported the performance of BBFlow if AlphaFold2-predicted structures, instead of the equilibrium structures from the ATLAS dataset, are passed to BBFlow. In this setting, we find only a slight decrease in performance compared to the equilibrium structures used in ATLAS, which can be expected since the MD simulations are not long enough (not converged to statistical equilibrium) for assuming independence of the starting structure. Importantly, BBFlow is still more accurate than the other baselines that have no access to the equilibrium structure from ATLAS, such as AlphaFlow without templates.
>
> While we regard AlphaFold2-predicted structures as the most relevant class of equilibrium structures in practice (AlphaFold2 is heavily used by practitioners and AlphaFold2-predicted structures are available for millions of naturally occurring proteins), we also ran an additional experiment during the rebuttal period, where we distort the equilibrium structures by adding noise.
> To this end, we added Gaussian noise to the Euclidean backbone coordinates of the ATLAS test set proteins and reran inference with BBFlow. We find that the performance remains strong and decreases only slightly. We report the results in Table R9 below.
>
> ||RMSF r|RMSF MAE|RMSF (MD=1.48)|pw-RMSD (MD=2.9)|pw-RMSD MAE|DCCM r|MD PCA W2|
> |:-|-:|-:|-:|-:|-:|-:|-:|
> |BBFlow|0.9|0.41|1.49|3.09|0.77|0.87|1.33|
> |BBFlow-0.1A|0.9|0.48|1.53|3.17|0.77 |0.87 |1.33|
> |BBFlow-0.2A|0.9 |0.48 |1.62 |3.3  |0.79 |0.87|1.32|
> |BBFlow-random conf as input |0.9 |0.47 |1.56|2.99|0.83|0.85|1.56|
>
> *Table R9. Performance of BBFlow on distorted equilibrium structures. We add Gaussian noise with the respective standard deviation to the backbone atoms equilibrium structure, or choose random MD conformations as equilibrium structure, and evaluate each model on the ATLAS test set. Units and settings as in Table 1.*
>
>
> &nbsp;
>
> *A3.4 - Addressing W3: Evaluation on short-timescale MD simulations, not on more complex dynamics*
>
> It is correct that also longer MD timescales and larger conformational changes are interesting. However, as we state in section 1.1, A.3, 'limitations' and 'Further baselines' in the original submission, this task is different from the one solved by AlphaFlow and BBFlow, in which MD is emulated under fixed, standardized settings. The ground truth ensembles strongly depend on these settings (temperature, timescale, forcefield). Thus, evaluating distributional accuracy quantitatively (and not rather qualitatively as in BioEmu) is only sensible if these settings are fixed. This becomes apparent in the poor performance of BioEmu on ATLAS as done in Table A.2 of the original submission and on de novo proteins in Table R10. This poor performance is expected because BioEmu does not claim to solve this specific task, conversely it can be expected that BBFlow and AlphaFlow would perform poorly if evaluated on BioEmu's task.
>
> Lastly, we would like to note that not only the prediction of far alternative states but also emulation of short-timescale MD under standardized setting has a lot of applications. Shese kind of MD simulations are employed on a regular basis by practitioners all around the world. To state a few concrete examples: MD on the 300 ns timescale can be used for assessing opening-closing movements, hinge movements or flexibility of binding pockets [1, 2, 3].
>
> || RMSF r | RMSF MAE | RMSF (MD=1.48) | pw-RMSD (MD=2.9) | pw-RMSD MAE | DCCM r | MD PCA W2 | Time/conf|
> |:--|--:|--:|--:|--:|--:|-:|--:|--:|
> |BioEmu|0.6|4.24 |7.56|7.9|8.29|0.64 |1.53 | 1.9|
> |BBFlow|**0.84**|**0.25**|**0.87**|**1.47**|**0.32**|**0.83**|**0.67**|**0.8**|
>
> *Table R10. Performance of BioEmu (and BBFlow) on the de novo proteins from Table 2 in the original submissions. Units and settings as in Table 2. Full set of metrics in Table R16.*
>
> ---
>
> [1] Raubenolt, Bryan A., et al. "Molecular dynamics simulations of the flexibility and inhibition of SARS-CoV-2 NSP 13 helicase." Journal of Molecular Graphics and Modelling 112 (2022)
>
> [2] Kiss, Gert et al. "Molecular dynamics simulations for the ranking, evaluation, and refinement of computationally designed proteins." Methods in Enzymology. Academic Press, 2013.
>
> [3] Wijma, Hein J. et al. "A computational library design protocol for rapid improvement of protein stability: FRESCO." Protein engineering: Methods and protocols, 2017. 69-85.
>
>
> &nbsp;
>
> *A3.5 - Addressing W4: No direct discussion or comparison with AlphaFlow-Lit*
>
> In AlphaFlow-LIT inference time is sped up by calculating the evoformer output only once. We refrained from including AlphaFlow-LIT as baseline because, although published as workshop paper at ICML 2024, no code, model weights or generated samples are publicly available. These would be required for evaluating the model. This would be necessary since AlphaFlow-LIT [1] does not report all metrics from AlphaFlow [2] and since in Table 1 of [1], inconsistent values for AlphaFlow are reported (see Table R11), raising the question whether the same metrics are being used by [1].
>
> We will include AlphaFlow-LIT as related work in the background section and explain, as above, why it cannot be evaluated.
>
> ||AF|AF in [1]|BBFlow|AF-Lit|
> |:-|-:|-:|-:|-:|
> |Pairwise CA-RMSD (=2.90)|2.89|2.89|3.09|2.43|
> |Pairwise CA-RMSD r|0.48|0.49|0.82|0.58|
> |MD CA RMSF|1.48|1.94|1.48|1.94|
> |CA RMSF|1.51|1.88|1.49|1.65|
> |Per target RMSF r|0.86|0.75|0.90|0.77|
> *Table R11. Performance of AlphaFlow-LIT as reported in [1]. There is an inconsistency between AlphaFlow (AF) as reported in [1] and as reported in [2] and our paper, respectively.*
>
> [1] Li S et al. Improving AlphaFlow for efficient protein ensembles generation. arXiv preprint arXiv:2407.12053, 2024.
> [2] B. Jing et al. AlphaFold Meets Flow Matching for Generating Protein Ensembles. ICML 2024
>
> &nbsp;
>
> *A3.6 - Addressing Q1: How sensitive is the model with respect to the hyperparameter xi?*
>
> The hyperparameter xi was chosen by brief, manual optimization on the validation set.
>
> We thank the reviewer for suggesting an ablation study, which we performed during the rebuttal period and report the results in Table R12. We observe that diversity, as measured by median RMSF and pairwise RMSD, increases with smaller xi, in the sense that xi=0.1 and xi=0.2 gives better diversity than xi=0.4. For the inference-xi-ablation (Table R7 in answer to QtLV), we observe a tradeoff between accuracy (RMSF r, RMSF MAE, PCA W2) and diversity.
>
> We also observe that training converges faster with larger xi, but we did not perform extensive analysis on this effect since it requires training several models per xi and we regard the importance of speeding up training as not too high, compared to speeding up inference.
>
> ||RMSF r|RMSF MAE|RMSF (MD=1.48)|pw-RMSD (MD=2.9)|pw-RMSD MAE|DCCM r|MD PCA W2|
> |:-|--:|-:|-:|--:|-:|-:|-:|
> | xi=0.1 |0.88 |0.54 |1.53 |2.74 |0.87 |0.86 |1.35|
> | xi=0.4 |0.89 |0.44 |1.39 |2.64 |0.89 |0.85 |1.33|
>
> *Table R12. Ablation of models trained with different choice for the hyperparameter xi. In the paper, we chose xi=0.2 based on manual optimization on the validation set. Units and settings as in Table 1.*
>
> &nbsp;
>
> *A3.7 - Addressing Q2: Why are metrics better for multi-chain systems reported in A.3?*
>
> We thank the reviewer for bringing this potential source of confusion to our attention. The reason for this effect is that some proteins are more challenging than others, thus the metrics between different sets of proteins cannot really be compared on an absolute scale, rather different baselines on the same set of proteins. Also, we would argue that there is no clear trend but it is rather mixed: RMSF r, RMSF median and DCCM r are worse than in Table 1 of the submission.

---

> > ### Comment · Reviewer_EssD · 2025-08-06
> >
> > I appreciate the authors’ efforts for rebuttal. While most of my concerns have been addressed, two issues remain and should be clarified in a revision:
> > * Technical contribution. Because BBFlow performs comparably to its closest counterpart, AlphaFlow-T, its primary contribution appears to be inference-time efficiency. This gain is modest, especially given the time-consuming nature of traditional MD simulations.
> > * Evaluation scope. The evaluation scenarios are limited. As noted, conformational variability over short-timescale MD simulations is small. The paper would benefit from additional results on proteins with complex, multi-state dynamics to demonstrate real-world utility.
> >
> > I would be happy to raise my score if these issues are further discussed.

---

> > > ### Author Response · Authors · 2025-08-06
> > >
> > > We thank the reviewer for their response to our rebuttal and for making us aware of the need to discuss the points raised in more detail. Below we discuss the points raised and explain corresponding remarks that we added to the final version of the paper.
> > >
> > > &nbsp;
> > >
> > > I) Technical contribution:
> > >
> > > The main technical contributions of BBFlow are as follows:
> > >
> > > (1) *Improved inference time-efficiency with little to no tradeoff in accuracy*.
> > >
> > > While other MD emulation baselines considered are either efficient or accurate, BBFlow does not tradeoff its increased time efficiency for less accurate predictions. We illustrate BBFlow’s favorable trade-off between efficiency and accuracy in Figure 4 in the original submission, where it visibly stands out.
> > >
> > > The efficiency gains are of significant magnitude: **The relative improvement of BBFlow over AlphaFlow is larger than the improvement of AlphaFlow over MD.**
> > > To provide evidence for the statement above, we will now estimate the inference time of MD per sampled conformation. The authors of AlphaFlow estimate the average MD inference time as 6.3 min/ns for ATLAS test set proteins (see Fig. 13 in AlphaFlow). Assuming that 1 ns of MD corresponds to a single state sampled with BBFlow or AlphaFlow-T, this MD achieves an inference time of around 400s per state.
> > > With 32s per state, **AlphaFlow is approximately 12 times faster than MD, while BBFlow is 500 times faster than MD and 40 times faster than AlphaFlow**. We thus disagree with the reviewer's characterization of BBFlow's gain in efficiency as 'modest, especially given the time-consuming nature of traditional MD simulations' and would appreciate it if the reviewer would reconsider their assessment of BBFlow's efficiency. For clarity, we added the above estimation also to the appendix of the final version.
> > >
> > > (2) *BBFlow is capable of predicting ensembles of multi-chain proteins while considered baselines are not*.
> > >
> > > As the reviewer (and all other reviewers) noted in their initial response as one of BBFlow’s pros, the model is capable of predicting ensembles of multi-chain protein structures, where it also achieves strong performance (Tab. A.3 and Figure 3 in the original submission). AlphaFlow, the most relevant baseline, fails to predict physical structures of a dimer (Fig. A.1). Also BioEmu does not support multi-chain systems.
> > >
> > > &nbsp;
> > >
> > > II) Evaluation scope is limited:
> > >
> > > As we explained in the answer A3.4 of the rebuttal, BBFlow is trained to emulate short-scale MD simulations provided by the ATLAS dataset. Long-timescale MD simulation is, as also in AlphaFlow [1], **out of scope of the paper** and stated as limitation already in the original submission. We thank the reviewer for making us aware of the need to discuss this in more detail and **will expand the limitations section of the final version further**, by discussing why sampling from **long-timescale MD ensembles is a separate task**, for which BBFlow is not designed.
> > >
> > > There we state that, similar to BioEmu's poor performance on short-timescale MD, we do not expect the BBFlow to perform well on long-timescale MD, i.e. to be capable of sampling large conformational changes that often involve folding transitions that occur on rather a micro- or millisecond timescale, as in multi-state systems.
> > > BioEmu has been designed to solve this task by training the model on private MD datasets spanning in total more than 200 milisecond trajectories, including folding transitions. These datasets are not publicly available to date.
> > > We show that BioEmu performs poorly if evaluated on short-timescale MD (Table A.5) and de-novo proteins (Table R10), providing evidence for the point that short-timescale and long-timescale ensemble generation are two separate tasks. There is currently no model that can solve both of these tasks at the same time.
> > >
> > > Even though BBFlow is expected to show poor performance on sampling far off-equilibrium states, we think that BBFlow can readily be used by the practitioners in real-world applications settings (as AlphaFlow-T, which is already being used), where short-timescale dynamics is being used for rational enzyme design and structural characterization of protein interactions. Several references supporting the importance of short-timescale dynamics can be found in answer A3.4. We will **discuss the applications of short-timescale MD in more detail in the final version**.
> > >
> > > We are happy to answer any further questions if needed.

---

> > > > ### Comment · Reviewer_EssD · 2025-08-07
> > > >
> > > > Thanks for the authors' response. I will consider raising the score.

---

### Official Review · Reviewer_QtLV · 2025-07-02

**Clarity:** 3
**Significance:** 3
**Originality:** 2
**Rating:** 4
**Confidence:** 4

**Summary:**

This work proposed BBFlow, a flow matching framework to generate conformational ensembles of protein. With conditional prior, the model is capable of generating high-fidelity conformational ensembles of protein backbones with short running time. The model can also generate conformational ensembles of multimers, which is valuable.

**Questions:**

1. A comment about the necessity of weights of a folding model or evolutionary sequence information: The structure itself contains sequence information implicitly. As BBFlow uses a prior that’s derived from the equilibrium structure, I believe it’s reasonable to expect the folding model weights or MSA is not critically needed. It is nice that the authors validated that through this paper.
1. It is not very fair to claim significant speedup over AlphaFlow because the BBFlow only generates the backbone ensemble instead of the all-atom ensemble. The computation burden of all-atom vs backbone generation is not comparable. This is my largest concern regarding this work.
1. Index encoding: Do authors have any intuition on why index encoding can hurt the performance of the model as shown in ablation? I understand the explanation in line 136-139, but that does not give intuition on the negative effect.
1. Ablation study: I am very curious about the ablation of the hyperparameter $\xi$ in the conditional prior. It would be nice if the author can show that.
1. Given conditional prior being close to the equilibrium state (or one of the many equilibrium states), I would imagine the model can better sample conformations that are close to the equilibrium state (geometrically or on the energy landscape). How does the model perform in terms of sample diversity? AlphaFlow offers some analysis on this front, and it would be nice if the authors can provide some analysis on sample diversity or energy landscape coverage.
1. Line 108: typo “benefitial”.

I'll be happy to raise my score if my concerns and questions are addressed.

**Ethical Concerns:**

["NO or VERY MINOR ethics concerns only"]

**Final Justification:**

Nice paper for protein backbone conformation ensemble generation that balances efficiency and accuracy. Despite the absence of side-chain related features and experiments, the work is valuable and should be accepted.

**Limitations:**

Yes

**Quality:**

3

**Strengths And Weaknesses:**

**Strength**
1. Interesting idea of backbone ensemble generation and well-executed experiments.
1. The proposed model BBFlow is capable of generating the ensemble of multimers, which is highly valuable.

**Weakness**
1. Inference time benchmark is not an apple-to-apple comparison. Baseline models such as AlphaFlow and ESMFlow generate protein conformation ensembles with all-atom resolution, while BBFlow only generate backbone ensembles.
1. Although the value of generating the backbone ensemble should be recognized, lack of side-chain details prohibited some important result analysis such as ramachandran plots and ensemble energy landscape.

---

> ### Author Rebuttal · Authors · 2025-07-30
>
> We cordially thank the reviewer for the time they invested in reading the paper and their constructive critique. We are happy that the reviewer finds our core ideas interesting, the experiments well-executed and highlights the valuable capability of BBFlow to sample ensembles of multimers!
> In the following, we address all points raised in the review. We will include the respective additional experiments and explanations in the final version of the paper.
>
>
> &nbsp;
>
> *A2.1 - Addressing W1/Q2: Inference time comparison is not fair*
>
> **Short answer:**
>
> This point is technically true, however, changing the settings to a strictly fair comparison only changes the values marginally. In the paragraph below, we provide a table with the respective values and remarks on the fairness of the comparison, which we also include in the final version of the paper. We thank the reviewer for bringing this potential misunderstanding to our attention, which allows us to clarify this in the final version as below.
>
> **Long answer:**
>
> AlphaFlow and ESMFlow first predict the backbone in the SE(3) frame representation, similar to BBFlow. As in AlphaFold2, the sidechain positions are then predicted in a post-processing step with a separate small neural network that only takes a fraction of the backbone-prediction inference time to run. Without considering sidechain conformations, AlphaFlow-T's inference time (per conformation on the protein 7c45A from Tab. 1 in original submission) is reduced from 32.6s to 32.5s while BBFlow only requires 0.8s. Thus, also in the strictly fair setting of backbone-only generation, BBFlow outperforms AlphaFlow-T in terms of efficiency, with virtually the same relative margin as reported in the original submission. In Table R5, we report the runtime of the individual modules of AlphaFlow and compare with BBFlow.
>
> | Module                               | Backbone | Sidechain |
> |--------------------------------------|----------|-----------|
> | AlphaFlow-T                          | 32.5 s    | 0.12 s     |
> | BBFlow                               | 0.8 s     | --        |
>
> *Table R5: Inference time by module of a single AlphaFlow and BBFlow forward pass on a 300 residue protein. The sidechain module is entirely separate from the backbone module and only takes a fraction of the time.*
>
> Lastly, we would like to note that AlphaFlow, during post-processing of the predicted backbone structure, uses the same sidechain module as AlphaFold2 without introducing methodological changes or ablating performance of the sidechain module. In AlphaFlow, as in our paper, the focus is on the backbone structure, which encodes larger conformational changes and the overall structure of the protein. If practitioners are interested in sidechain configurations of the generated backbone ensembles, it is, in principle, possible to combine the sidechain module from AlphaFold2 also with BBFlow, as done in AlphaFlow. However, this requires re-training and is out of scope for the paper at hand.
>
>
> &nbsp;
>
> *A2.2 - Addressing W2: Lack of Ramachandran analyses and sidechain energy landscapes*
>
> We added Ramachandran histograms of backbone atom dihedrals to the paper, comparing BBFlow, AlphaFlow-T and MD for four randomly sampled proteins from ATLAS.
> We find that the Ramachandran histograms of both BBFlow and AlphaFlow-T indeed show overlap with MD (we cannot upload the plots during rebuttal since the global response with PDF is not supported this year and links are not allowed).
> The Ramachandran plots are possible because backbone dihedrals can be calculated from BBFlow's predicted Euclidean coordinates for the backbone atoms. For energy landscapes, however, the sidechain conformation is required in order to apply a force field.
> While further sidechain analyses would be interesting indeed, they are out of scope for both our paper and also AlphaFlow. Instead, the main contribution of both papers is predicting ensembles of backbone structures. The backbone structure captures overall conformational changes and dominates computational cost of state-of-the-art models like AlphaFold2 or Alphaflow, as shown above. Thus, also the authors of AlphaFlow (the most relevant baseline) do not perform analyses on the recovery of the sidechain conformations, Ramachandran plots or energy landscapes, possibly because the sidechain prediction method is not novel but done as post-processing step with the same network as in AlphaFold2, as described above.
>
>
> &nbsp;
>
> *A2.3 - Addressing Q4: Ablation study on the xi hyperparameter*
>
> We thank the reviewer for suggesting this experiment, which we performed during the rebuttal period and report the results in Table R6. We observe that diversity, as measured by median RMSF and pairwise RMSD, increases with smaller xi, in the sense that xi=0.1 and xi=0.2 gives better diversity than xi=0.4. For the inference-xi-ablation (Table R7), we observe a tradeoff between accuracy (RMSF r, RMSF MAE, PCA W2) and diversity.
>
> We also observe that training converges faster with larger xi, but we did not perform extensive analysis on this effect since it requires training several models per xi and we regard the importance of speeding up training as not too high, compared to speeding up inference.
>
>
> ||RMSF r|RMSF MAE|RMSF (MD=1.48)|pw-RMSD (MD=2.9)|pw-RMSD MAE|DCCM r|MD PCA W2|
> |:-|--:|-:|-:|--:|-:|-:|-:|
> | xi=0.1 |0.88 |0.54 |1.53 |2.74 |0.87 |0.86 |1.35|
> | xi=0.4 |0.89 |0.44 |1.39 |2.64 |0.89 |0.85 |1.33|
>
> *Table R6. Ablation of models trained with different choice for the hyperparameter xi. In the paper, we chose xi=0.2 based on manual optimization on the validation set. Units and settings as in Table 1.*
>
>
> | xi   | RMSF r | RMSF MAE | RMSF (MD=1.48) | pw-RMSD (MD=2.9) | pw-RMSD MAE | DCCM r | MD PCA W2 |
> |-----:|-------:|---------:|---------------:|-----------------:|------------:|-------:|----------:|
> | 0.01 |   0.7  |     7.61 |          11.16 |            14.15 |       10.83 |   0.73 |      2.85 |
> | 0.05 |   0.81 |     3.89 |           5.43 |             8.19 |        5.38 |   0.78 |      2.06 |
> | 0.1  |   0.88 |     1.32 |           2.62 |             5.2  |        2.21 |   0.84 |      1.51 |
> | 0.2  |   0.9  |     0.41 |           1.49 |             3.09 |        0.77 |   0.87 |      1.33 |
> | 0.3  |   0.89 |     0.47 |           1.37 |             2.49 |        1.02 |   0.86 |      1.69 |
> | 0.4  |   0.86 |     0.64 |           1.65 |             2.5  |        1.31 |   0.8  |      2.12 |
> | 0.5  |   0.79 |     0.8  |           1.82 |             2.6  |        1.5  |   0.74 |      3    |
> | 0.6  |   0.73 |     0.82 |           1.7  |             2.33 |        1.65 |   0.7  |      4.51 |
>
> *Table R7. Ablation of the model from the paper, trained with xi=0.2, for inference with different values of xi. Units and settings as in Table 1.*
>
>
> &nbsp;
>
> *A2.4 - Addressing Q5: Analysis on sample diversity*
>
> We can understand the concern of the reviewer that BBFlow might be biased towards predicting states that are close to the equilibrium structure since our method heavily relies on that structure. However, already from the median RMSF column reported in Tables 1 and 2 from the original submission, it can be seen that BBFlow predicts off-equilibrium states to a similar extent as MD, while AlphaFlow-T samples too close to equilibrium, as indicated by the smaller median RMSF. We will stress this important point more in the final version.
>
> To make the observation more explicit, we introduced an additional analysis during the rebuttal period. We calculated the average RMSD of the generated samples from the equilibrium structure for each protein and report the results in Table R8 below. We find the same trend as in the original submission: BBFlow samples states that are on average as far away from the equilibrium structure as those explored by MD, while AlphaFlow-T samples states that are too close to the equilibrium structure.
>
> | Method       | Mean RMSD to Eq. Structure [Å] |
> |--------------|--------------------------|
> | MD           | 3.0                      |
> | BBFlow       | **2.5**                  |
> | AlphaFlow-T  | 2.1                      |
> | BioEMU       | 4.0                      |
> | AlphaFlow    | **3.5**                  |
>
> *Table R8. Deviation from the equilibrium structure on the ATLAS test set. Closer to MD is better.*
>
>
> &nbsp;
>
> *A2.5 - Addressing Q3: Intuition for removal of index encoding*
>
> Intuitively, removing the index encoding reduces the potential for overfitting and incentivizes the model to generalize better. If the index of a certain (challenging) structural motif in the training set is available, the model might use the combination of motif and index in order to memorize the dynamics in the training set. The index should have no physical meaning but only structural context and amino acid identity, removing it as input can thus be interpreted as physical inductive bias. We will add a comment on this to the final version and thank the reviewer for making us aware of the need for clarification.
>
>
> &nbsp;
>
> *A2.6 - Addressing Q1: Comment about the necessity of weights of a folding model or evolutionary information*
>
> We agree with the reviewer and identify this as one of the core ideas that underlie our method. We would like to note that also all other reviewers find the idea 'elegant', 'technically compelling' or 'conceptually reasonable'.
>
> &nbsp;
>
> *A2.7 - Addressing Q6: typo*
>
> We corrected the typo and thank the reviewer for pointing this out!

---

### Official Review · Reviewer_aKex · 2025-07-03

**Clarity:** 3
**Significance:** 3
**Originality:** 3
**Rating:** 5
**Confidence:** 3

**Summary:**

In this paper, the authors present a method called "BBflow" for sampling protein structures from MD-derived steady state ensembles using a flow matching framework. The key idea is to condition on a template structure (the "equilibrium structure") in order to avoid the problem of requiring the neural network to be able to predict the general structure from sequence before learning ensemble conformations. As a result, the author's model can be trained from scratch rather than building on a pre-trained structure predictor and does not require multiple sequence alignments as input.

BBflow is trained on the Atlas dataset and is primarily compared with AlphaFlow, where BBflow shows similar to favorable metrics. The authors also consider de novo designed proteins and multi chain proteins which are challenging or impossible for current methods.

**Questions:**

* How sensitive is the model to the quality of the equilibrium structure and can samples escape the energy well around that structure?
* Similarly, for conditional prior distribution, does the change of xi affect the training time, e.g., if we have a prior closer to the equilibrium structure, the model converges faster? But then, will this change the diversity of the predicted conformations maybe?
* In principle, evolutionary information should encode protein dynamics if those dynamics are important for function/fitness. Are there situations where excluding this information is harmful?

**Ethical Concerns:**

["NO or VERY MINOR ethics concerns only"]

**Final Justification:**

The rebuttal mostly addressed my questions, but I think this is a useful contribution anyway and is high enough quality for acceptance.

**Limitations:**

yes

**Quality:**

3

**Strengths And Weaknesses:**

Strengths
* The paper is well written, the method and results are clearly described, and the method is well motivated
* The core idea is elegant (condition on the equilibrium structure and let the model just learn the physics rather than needing to also fold the structure from scratch) and its merit bears out in the experiments
* Experiments are comprehensive

Weaknesses
* This is not specific to this work, but the metrics used (e.g. in Table 1) are not intuitive. It's hard to understand how exactly these correspond to, e.g., whether the densities of the learned distribution match the ground truth and what good baselines would be. For example, pairwise RMSD is influenced by the actual diversity of the MD samples (if the MD samples are high diversity, pairwise RMSD should be worse) but we don't have an MD replicate baseline to understand how far these methods are from "perfect"
* Not clear how sensitive the method is to poorly estimated equilibrium structures and the method seems pretty reliant on these being accurate. Can it effectively escape the well of the equilibrium structure to sample other low energy conformations or does it primarily stay around the equilibrium structure? Looking at a protein with known large conformational shifts would be interesting
* Some more details on the implementation/algorithms for the components used from existing works (although cited) would make the method easier to follow as a stand alone work

---

> ### Author Rebuttal · Authors · 2025-07-30
>
> We cordially thank the reviewer for their feedback on our paper. We are happy to read the reviewer finds our paper well-written, the core idea elegant and the experiments comprehensive. Below, we address these points raised in the review and present corresponding additional experiments that we ran during the rebuttal period.
>
> &nbsp;
>
> *A1.1 - Addressing W2/Q1: Sensitivity of the model to the quality of the equilibrium structure*
>
> **Short answer:**
>
> If the quality of the equilibrium structure is reduced, BBFlow's performance only decreases slightly, indicating robustness under noisy equilibrium structures.
>
> **Long answer:**
>
> We agree with the reviewer that robustness under changes of the equilibrium structure is a good consistency check and also relevant in practice. In Table A.1 of the original submission, we reported the performance of BBFlow if AlphaFold2-predicted structures, instead of the equilibrium structures from the ATLAS dataset, are passed to BBFlow. In this setting, we find only a slight decrease in performance compared to the equilibrium structures used in ATLAS, which can be expected since the MD simulations are not long enough (not converged to statistical equilibrium) for assuming independence from the starting structure. Importantly, BBFlow is still more accurate than the other baselines that have no access to the equilibrium structure from ATLAS, such as AlphaFlow without templates.
>
> While we regard AlphaFold2-predicted structures as the most relevant class of equilibrium structures in practice (AlphaFold2 is heavily used by practitioners and AlpaFold2-predicted structures are available for millions of naturally occurring proteins), we also ran an additional experiment during the rebuttal period, where we distort the equilibrium structures by adding noise.
> To this end, we added Gaussian noise to the Euclidean backbone coordinates of the ATLAS test set proteins and reran inference with BBFlow. We find that the performance remains strong and decreases only slightly. We report the results in Table R1 below.
>
> ||RMSF r|RMSF MAE|RMSF (MD=1.48)|pw-RMSD (MD=2.9)|pw-RMSD MAE|DCCM r|MD PCA W2|
> |:-|-:|-:|-:|-:|-:|-:|-:|
> |BBFlow|0.9|0.41|1.49|3.09|0.77|0.87|1.33|
> |BBFlow-0.1A|0.9|0.48|1.53|3.17|0.77 |0.87 |1.33|
> |BBFlow-0.2A|0.9 |0.48 |1.62 |3.3  |0.79 |0.87|1.32|
> |BBFlow-random conf as input |0.9 |0.47 |1.56|2.99|0.83|0.85|1.56|
>
> *Table R1. Performance of BBFlow on distorted equilibrium structures. We add Gaussian noise with the respective standard deviation to the backbone atoms equilibrium structure, or choose random MD conformations as equilibrium structure, and evaluate each model on the ATLAS test set. Units and settings as in Table 1.*
>
> &nbsp;
>
> *A1.2 - Addressing W2/Q1: Escaping the energy well around the eq. structure*
>
> We can understand the concern of the reviewer that BBFlow might be biased towards predicting states that are close to the equilibrium structure since our method heavily relies on that structure. However, already from the median RMSF column reported in Tables 1 and 2 from the original submission, it can be seen that BBFlow predicts off-equilibrium states to a similar extent as MD, while AlphaFlow-T samples too close to equilibrium, as indicated by the smaller median RMSF. We will stress this important point more in the final version.
>
> To make the observation more explicit, we introduced an additional analysis during the rebuttal period. We calculated the average RMSD of the generated samples from the equilibrium structure for each protein and report the results in Table R2 below. We find the same trend as in the original submission: BBFlow samples states that are on average as far away from the equilibrium structure as those explored by MD, while AlphaFlow-T samples states that are closer to the equilibrium structure.
>
>
> | Method       | Mean RMSD to Eq. Structure [Å] |
> |--------------|--------------------------|
> | MD           | 3.0                      |
> | BBFlow       | **2.5**                  |
> | AlphaFlow-T  | 2.1                      |
> | BioEMU       | 4.0                      |
> | AlphaFlow    | **3.5**                  |
>
> *Table R2. Deviation from the equilibrium structure on the ATLAS test set. Closer to MD is better.*
>
> &nbsp;
>
> *A1.3 - Addressing Q2: Effect of hyperparameter xi on training time and diversity of ensembles*
>
> We thank the reviewer for suggesting this experiment, demonstrating the effect of xi, which we performed during the rebuttal period and report the results in Table R3. We indeed observe that diversity increases with smaller xi, in the sense that pairwise RMSD and median RMSF of the model trained with xi=0.4 is smaller than that of the xi=0.2 and xi=0.1 models. For the inference-xi-ablation (Table R4), we observe a tradeoff between accuracy (RMSF r, RMSF MAE, PCA W2) and diversity.
>
> As anticipated by the reviewer, we also observe that training converges faster with larger xi, but we did not perform extensive analysis on this effect since it requires training several models per xi and we regard the importance of speeding up training as not too high, compared to speeding up inference.
>
> ||RMSF r|RMSF MAE|RMSF (MD=1.48)|pw-RMSD (MD=2.9)|pw-RMSD MAE|DCCM r|MD PCA W2|
> |:-|--:|-:|-:|--:|-:|-:|-:|
> | xi=0.1 |0.88 |0.54 |1.53 |2.74 |0.87 |0.86 |1.35|
> | xi=0.4 |0.89 |0.44 |1.39 |2.64 |0.89 |0.85 |1.33|
>
> *Table R3. Ablation of models trained with different choice for the hyperparameter xi. In the paper, we chose xi=0.2 based on manual optimization on the validation set. Units and settings as in Table 1.*
>
> | xi   | RMSF r | RMSF MAE | RMSF (MD=1.48) | pw-RMSD (MD=2.9) | pw-RMSD MAE | DCCM r | MD PCA W2 |
> |-----:|-------:|---------:|---------------:|-----------------:|------------:|-------:|----------:|
> | 0.01 |   0.7  |     7.61 |          11.16 |            14.15 |       10.83 |   0.73 |      2.85 |
> | 0.05 |   0.81 |     3.89 |           5.43 |             8.19 |        5.38 |   0.78 |      2.06 |
> | 0.1  |   0.88 |     1.32 |           2.62 |             5.2  |        2.21 |   0.84 |      1.51 |
> | 0.2  |   0.9  |     0.41 |           1.49 |             3.09 |        0.77 |   0.87 |      1.33 |
> | 0.3  |   0.89 |     0.47 |           1.37 |             2.49 |        1.02 |   0.86 |      1.69 |
> | 0.4  |   0.86 |     0.64 |           1.65 |             2.5  |        1.31 |   0.8  |      2.12 |
> | 0.5  |   0.79 |     0.8  |           1.82 |             2.6  |        1.5  |   0.74 |      3    |
> | 0.6  |   0.73 |     0.82 |           1.7  |             2.33 |        1.65 |   0.7  |      4.51 |
>
> *Table R4. Ablation of the model from the paper, trained with xi=0.2, for inference with different values of xi. Units and settings as in Table 1.*
>
> &nbsp;
>
> *A1.4 - Addressing Q3: Evolutionary information encoding dynamics via fitness*
>
> Indeed, evolutionary information (e.g. MSA) can encode information about functionally or structurally important sites in proteins and their dynamics.
>
> As we demonstrate in our paper, evolutionary information is not strictly required for emulation of 300ns MD simulation if abundant training data (such as the ATLAS dataset) is present. We expect that evolutionary information might become a necessity for predicting conformational changes occurring at longer timescales, where the task becomes increasingly more challenging, especially given the lack of large datasets covering micro- and millisecond protein dynamics. This is, however, out of scope of the paper as we describe in sections 1.1, A.3 and 'limitations' of the original submission. We added the above discussion on the role of evolutionary information to section A.3.
>
> &nbsp;
>
> *A1.5 - Addressing W3: More detailed explanations on models and algorithms cited from other papers*
>
> We thank the reviewer for making us aware of this. We will add an overview on the respective resources to the appendix of the final version.

---

> > ### Comment · Reviewer_aKex · 2025-08-07
> >
> > Thanks for your response. I remain positive about this paper, think it is a useful contribution to the community, and maintain my accept recommendation.
> >
> > I appreciate the additional experimental results and analyses. Regarding the equilibrium structure experiments, it's worth noting that, due to the nature of your test set, the AF2 equilibrium structure predictions are likely better than they would be for a random protein not in ATLAS/the PDB. Adding Gaussian noise or sampling at random from the MD conformations likely doesn't reflect the same magnitude of error we would see for a genuine holdout protein. That said, this is not exactly a unique problem to this method, but it would be interesting to see what happens with proteins with large domain motions or switch like structural changes. Will the alternative state be present in the generated ensemble if the equilibrium structure comes from one structural mode or the other?

---

> > > ### Author Response · Authors · 2025-08-07
> > >
> > > Thank you for the response to our rebuttal. We are happy to read that you remain positive about our paper.
> > >
> > > We agree that investigating state-switching proteins with respect to the equilibrium structure and out-of-distribution proteins would be an interesting direction for future work. We would like to note that the de-novo proteins investigated in our paper can be considered out-of-distribution to some extent. Here we find that BBFlow and AlphaFlow-T outperform methods that do not rely on the equilibrium structure but rather evolutionary information, such as BioEmu (Table R10) and AlphaFlow without templates (Table 2 in the paper).
> > >
> > > We are happy to answer any further questions if needed!

---

### Note · Authors · 2025-08-13

During the rebuttal, all concerns could be resolved and no final questions remained. We would thus like to use the final comment to summarize the rebuttal.

&nbsp;

We thank all reviewers for their constructive comments and questions. We are pleased that the reviewers find our idea elegant and interesting (**aKex**, **QtLV**), the method compelling (**EssD**), and the experiments comprehensive and well-executed (**aKex**, **QtLV**), fully supporting the main claims of our paper (**ndHw**). The reviewers highlight the novel capability of the model to generate ensembles of multimers (**QtLV**, **EssD**) and de novo proteins (**aKex**), and the improvement in inference time (**ndHw**). During the rebuttal, we were able to resolve the concerns raised by reviewers. Below, we summarize the main concerns.

(1) Inference time efficiency *(concern of QtLV, EssD)*: In a direct comparison to AlphaFlow, BBFlow's improvement in terms of inference time is of significant magnitude **(see answer A2.1, Table R5)**, which is also meaningful in practice: BBFlow is around 40 times faster than AlphaFlow, while AlphaFlow is only around 12 times faster than the ground truth, Molecular Dynamics, in a representative setting **(see final comment to EssD)**.

(2) Sensitivity of BBFlow to noisy equilibrium structures *(concern of aKex, EssD)*: We ran an additional experiment and found that with reduced quality of the input equilibrium structures, BBFlow's performance only decreases slightly **(answer A1.1, Table R1)**.

(3) Additional performance metrics, comparison with BioEmu *(concern of ndHw)*: We updated the submission text and included additional metrics, such as weak contact metrics, in the main text **(see answer A4.2)**. As suggested by the reviewer, we included BioEmu's performance in the main text and stressed that BBFlow outperforms BioEmu in all metrics **(see answer A4.3)**. We also explained why a direct comparison to BioEmu is problematic.

We also performed other additional experiments such as the ablation of the hyperparameter xi **(see answer A1.3)**.


We think that the additional experiments and clarifications not only resolve prior concerns but also further strengthen the quality and impact of our submission, for which we, again, would like to thank the reviewers.

---

### Decision · Program_Chairs · 2025-09-17

**Decision:**

Accept (poster)

**Comment:**

This paper introduces BBFlow which generates samples from the Boltzmann distribution by conditioning on a starting equilibrium structure in the model as well as modifying the prior. All reviewers found the idea sensible, and the performance gains over existing approaches like AlphaFlow are appreciated. A few reviewers questioned the inference speedup over AlphaFlow, but the authors included additional details in their rebuttal responses to clarify this aspect. Overall, the proposed idea is empirically interesting and well executed, but does not provide a lot of technical novelty. As a result, I recommend acceptance as a poster.